

# Small scale variability of geomorphological settings influences mangrove-derived organic matter export in a tropical bay

Geraldina Signa[a,*], Antonio Mazzola[a], James Kairo[b], Salvatrice Vizzini[a]

[a] Department of Earth and Marine Sciences, University of Palermo, CoNISMa, via Archirafi 18, Palermo, Italy

[b] Kenya Marine and Fisheries Research Institute, PO Box 81651, Mombasa, Kenya

*Correspondence to:* Geraldina Signa (geraldina.signa@unipa.it)

## Abstract

Organic matter (OM) exchanges between adjacent habitats affect the dynamics and functioning of coastal systems, as well as the role of the different primary producers as energy and nutrient sources in food webs. Elemental (C, N, C:N) and isotope ($\delta^{13}C$) signatures and fatty acid (FA) profiles were used to assess the influence of geomorphological setting in two climatic seasons on the export and fate of mangrove OM across a tidally influenced tropical area, Gazi Bay (Kenya). The main results indicate that tidal transport, along with riverine runoff, play a significant role in the distribution of mangrove organic matter. In particular, a marked spatial variability in the export of organic matter from mangroves to adjacent habitats was due to the different settings of the creeks flowing into the bay. Kinondo Creek acted as a mangrove retention site, where export of mangrove material was limited to the contiguous intertidal area, while Kidogoweni Creek acted as a "flow-through" system, from which mangrove material spread into the bay, especially in the rainy season. This pattern was evident from the isotopic signature of primary producers, which were more $^{13}C$-depleted in the Kinondo Creek and nearby, due to the lower dilution of the DIC pool, typically depleted as an effect of intense mangrove mineralization. Despite the trapping efficiency of the seagrass canopy, suspended particulate OM showed the important contribution of mangroves across the whole bay, up to the coral reef, as an effect of the strong ebb tide. Overall, mixing model outcomes and FA profiles indicated a widespread mixed contribution of both allochthonous and autochthonous OM sources across Gazi Bay. Moreover, FAs indicated a notable contribution of brown macroalgae and





bacteria in both sediment and particulate pools. These results suggest that ecological connectivity in Gazi Bay is strongly influenced by geomorphological setting, which may have far-reaching consequences for the functioning of the whole ecosystem and the local food webs.

# 1    Introduction

Mangrove forests are known to be among the most productive ecosystems worldwide, with a crucial role in the carbon budget (Alongi, 2014; Clough, 1998; Dittmar et al., 2006; Kristensen et al., 2008). Indeed, mangroves sequester large amounts of carbon deriving from two main pathways: atmospheric carbon fixation contributing to high aboveground and belowground biomass and carbon input from adjacent terrestrial and marine systems (Alongi, 2014). However, a considerably large amount of mangrove-derived carbon in the form of leaves, particulate detritus and dissolved organic and inorganic matter is exported to adjoining ecosystems, subsidising coastal waters and influencing nutrient biogeochemical cycling (Dittmar et al., 2006; Duarte and Cebrián, 1996).

The long-standing scientific debate regarding the fate of the high mangrove productivity has led to various theories and paradigms. With the "outwelling hypothesis", Odum and Heald (1972) first suggested that leaf detritus exported from mangrove forests would represent an important trophic source contributing to secondary production in adjacent offshore areas. While a few studies confirmed this hypothesis (Dittmar et al., 2001; Lugendo et al., 2007), others reported limited evidence that exported mangrove detritus subsidises offshore food webs (Lee, 1995). More recently, a tight interlinkage between mangroves and adjacent seagrasses has been found in many geographical areas (e.g. Hemminga et al. 1994, Bouillon et al. 2003, 2007, Kennedy et al. 2004, Gillis et al. 2014), leading to the "inwelling hypothesis", whereby the tidal import of seagrass litter to mangrove forests and its contribution to the carbon budget would be also significant (Bouillon et al., 2003; Walton et al., 2014).

According to the "environmental setting hypothesis" (Twilley, 1985), geomorphological features and hydrology are the most important abiotic factors affecting the exchange of mangrove material across ecosystem boundaries (Adame and Lovelock, 2011; Lee, 1999). In particular, the magnitude and frequency of river discharge, tidal amplitude, rainfall and wave power are reported to influence the exchange of material across boundaries. Tropical rivers are one of the main vectors of carbon from



terrestrial to coastal areas (Bouillon and Connolly, 2009), especially when the flushing rate is amplified by high rainfall (Dittmar et al., 2001). Otherwise, low rainfall and freshwater runoff have been related to reduced mangrove outwelling (Walton et al., 2014). Tidal amplitude, frequency and direction (flood *vs.* ebb tide) influence cross-habitat connectivity in the tropical coastal seascape, through bidirectional

movements of water and nekton across wide spatial scales (Adame and Lovelock, 2011).

Bulk elemental composition (C, N and C:N ratio) and carbon stable isotope (SI) signature ($\delta^{13}$C) are widely used to trace organic matter origin and fate in coastal systems (e.g. Bouillon et al. 2003, Kennedy et al. 2004, Walton et al. 2014). This is because C:N ratio and $\delta^{13}$C composition of the different organic matter sources contributing to sedimentary and suspended particulate pools are often markedly distinct

(Bouillon et al., 2008; Lamb et al., 2006). Nevertheless, this dual approach has a number of constraints, such as the lowering effect of decomposition on C:N ratio (Bosire et al., 2005; Dunn et al., 2008; Kennedy et al., 2004) or the potential overlapping of basal source isotopic signatures (Fry and Sherr, 1989), which may complicate inferences in coastal areas where autochthonous and allochthonous material mix. The simultaneous use of other complementary tracers, such as fatty acids (FAs), may significantly improve

the determination of OM sources in aquatic systems (Alfaro et al., 2006; Dunn et al., 2008) and their trophic pathways, both detrital and grazing pathways, in complex systems (Kelly and Scheibling, 2012). Indeed, FAs are particularly well-suited to distinguishing among a wide range of organic matter sources (i.e. dinoflagellates, diatoms, macroalgae, seagrasses, bacteria) because of their high biological specificity and their structural and trophic stability (Kelly and Scheibling, 2012; Meziane et al., 2006).

Gazi Bay in Kenya is a tropical area characterised by high cross-habitat contiguity. The bay is fringed by a dense mangrove forest and the seabed covered with seagrass beds mixed with patchy macroalgae. Two tidal creeks, of which only one receives riverine freshwater input, intersect the mangrove forest. Rainfall seasonality exerts a marked influence on runoff magnitude. Earlier studies in Gazi Bay reported a marked outwelling of mangrove-derived material up to the adjacent seagrass beds (Bouillon et al., 2007;

Hemminga et al., 1994) and also a reversed inwelling of organic seagrass material that contributes to a tight linkage between these two contiguous habitats (Hemminga et al., 1994). In contrast, adjacent coral reefs have been found to be fairly isolated from the inshore habitat (Hemminga et al., 1994). However, previous investigations were limited to only a few parts of the bay or only one climatic season, and were



conducted exclusively through a qualitative analysis of stable isotopes. Moreover, only tidal pumping was considered as the main driver of outwelling and inter-habitat connectivity. In contrast, the role of small-scale variability in the geomorphology of the bay, in terms of tidal creek length, depth and freshwater runoff on mangrove export and cross-habitat organic matter exchange, has never been

assessed. Given the importance and ecological implications of mangrove export along the river-bay-coral reef-open sea continuum, a higher spatial resolution, combined with a conjoint analytical approach and a quantitative data elaboration, provides more detailed and complete quantitative information.

The main goal of this study was to assess the fate of mangrove-derived organic matter in Gazi Bay, in relation to the small-scale variability of geomorphological settings, using a combined elemental, isotopic

and fatty acid approach. In particular, we assessed: a) export towards adjacent habitats in relation to the different geomorphological settings of the bay in the two main climatic seasons, dry and rainy; b) the contribution of dominant primary producers to sedimentary and suspended particulate organic matter pools.

## 2      Materials and methods

### 2.1      Study area and sample collection

Gazi Bay (4°25'S; 39°50'E) is a ~10 km$^2$ shallow semi-enclosed marine ecosystem on the southern Kenyan coast (Fig. 1). In the northern part it is fringed by a ~6 km$^2$ mangrove forest, while in the southern part it is sheltered from the Indian Ocean by shallow coral reefs. Kidogoweni and Kinondo tidal creeks

flow through the mangrove forest into the upper part of the bay. Kinondo Creek is shorter and shallower than Kidogoweni (2500 m long and 1-2 m deep *vs.* 4500 m long and 2-3 m deep) and lacks direct freshwater input, unless low groundwater discharge (Kitheka et al., 1996). Kidogoweni Creek receives freshwater input from the semiperennial Kidogoweni River. A third freshwater input, the Mkurumuji River, discharges into the southern part of the bay with high flow rates (Kitheka et al., 1996). The tide is

semidiurnal, ranging from 0.5 m at neap tide to 4.0 m at high spring tide (Kitheka et al., 1996). Tidal currents are faster in the creeks than in the southern part of the bay and ebb flows are stronger than flood ones (Kitheka, 1997). Two main climatic seasons and a bimodal rainfall distribution characterise the area:



a distinct dry season (January– February) is followed by a long and intense rainy season (April–August) and a shorter and weaker one (October-November).

The dominant mangrove species along the creek fringes at Gazi bay are *Rhizophora mucronata, Ceriops tagal, Sonneratia alba* and *Avicennia marina* (Hemminga et al., 1994; Neukermans et al., 2008). Seagrass

beds mixed with macroalgae cover both the creeks and the bay up to the seaward mouth (Githaiga et al., 2016). *Thalassodendron ciliatum, Cymodocea serrulata, C. rotundata* and *Enalhus acoroides* are among the most abundant seagrass species, while the brown macroalgae *Sargassum binderi, Dictyota cervicornis, Turbinaria conoides* and the red ones *Gracilaria corticata* and *Hypnea cornuta* are among the most abundant macroalgae.

Sampling was carried out in February and July 2009, during the dry and rainy seasons respectively, in four stations: mangrove forest (M), intertidal area (IA), seagrass bed (SB) and coral reef (CR), situated along two land-to-sea transects, corresponding to the two tidal creek mouths: Kidogoweni (A) and Kinondo (B) (Fig. 1). In both seasons, sampling was carried out at spring tides. At each station, three replicates of surface water and sediment were collected using 5*l* bottles and hand corers (length: 20 cm;

inner diameter: 4 cm) respectively. The most abundant organic matter sources (senescent yellow mangrove leaves, seagrass leaves and macroalgae) were collected by hand. Zooplankton were captured towing a net (335 μm mesh) for 15 minutes at constant speed in each station. All samples were kept cool and dark upon arrival at the laboratory.

## 2.2    Laboratory activities and analytical procedures

Surface water was pre-filtered using a 200 μm net and then filtered onto precombusted (450°C, 4 h) Whatman GF/F filters to collect suspended organic matter (SPOM). The first centimetres (∼5 cm) of sediment cores were sliced, homogenized and subsampled for subsequent analysis of sedimentary organic matter (SOM). After species identification, mangrove and seagrass leaves and macroalgae were quickly

rinsed to remove any extraneous material, and, when present, epiphytes were scraped. Zooplankton samples were sorted under a stereo microscope (10 to 40×). Subsamples for elemental and isotopic analyses were dried at 60°C to constant weight and ground to a fine powder using a micro-mill. Filters and sediments for carbon stable isotope analysis were drop-by-drop acidified (HCl, 2 N) to remove



carbonates, washed and dried again. Subsamples for fatty acid analysis were frozen (-80°C), freeze-dried (ALPHA 1-4 LD plus, Martin-Christ) and ground into a powder using a micro-mill. Ground samples were kept at -80 °C until analysis.

An elemental analyser (Thermo Flash EA 1112) was used for the determination of total carbon and nitrogen, and connected to an isotope ratio mass spectrometer (Thermo Delta Plus XP) for $\delta^{13}C$ analysis. $\delta^{13}C$ was expressed in relation to PeeDee Belemnite international standard, and defined as: $\delta^{13}C = [(R_{sample}/R_{standard})-1] \times 10^3$, where R is the $^{13}C/^{12}C$ ratio. Analytical precision based on the standard deviation of replicates of internal standards (International Atomic Energy Agency IAEA-CH-6) was 0.2 ‰.

Lipids were extracted from suspended particulate organic matter collected in filters, surface sediment and primary producers using a distilled water/methanol/chloroform solvent mixture, following a modified version of the Bligh and Dyer (1959) method. A 0.01% solution of butylated hydroxyl toluene (BHT) was added to prevent lipid oxidation. Lipid extract was subjected to acid-catalysed transesterification with methanolic hydrogen chloride to obtain fatty acids (FAs). FAs were then analysed as methyl esters (FAME) by gas chromatography (GC-2010, Shimadzu) equipped with a BPX-70 capillary column (30 m length; 0.25 mm ID; 0.25 µm film thickness) and with a flame ionization detector (FID). Peaks were identified using retention times from mixed commercial standards (37FAME and BAME from Supelco; BR1 and QUALFISH from Larodan). For quantification, tricosanoic acid (C23:00) was used as internal standard. Individual FAs were expressed as percentage of total FAs.

## 2.3   Data analysis

Bayesian mixing models were run on $\delta^{13}C$ data using the software package SIAR (Stable Isotope Analysis in R) (Parnell et al., 2010) to estimate the contribution of each organic matter source to sedimentary and suspended particulate organic matter. No trophic enrichment factor was included in the models. Mangroves (Mang), macroalgae (Alg) and seagrasses (Seag) were chosen as end-members for sedimentary organic matter. Oceanic SPOM was not included in the model as a proxy of phytoplankton because of the small proportion of phytoplankton biomass within the POM pool in Gazi Bay (Bouillon et



al., 2007). Mangroves (Mang), macroalgae (Alg), seagrasses (Seag) and zooplankton (Zoo) were chosen as end-members for suspended particulate organic matter.

Multivariate analysis was performed on FA data using Primer 6 & Permanova packages (Plymouth Marine Laboratory). FA data were resembled using Bray Curtis similarity after arcsine square root function transformation and permutational multivariate analysis of variance (PERMANOVA) was carried out to examine species differences within each primary producer group (mangroves, red and brown macroalgae, seagrasses). Non-metric multidimensional scaling (nMDS) ordination was carried out to compare FA profiles of the different primary producer groups. PERMANOVA was then performed to test differences among primary producer groups. The analysis of percentage similarity (SIMPER) was used to identify which FAs could be used as indicators of primary producer-derived organic matter in sedimentary and suspended particulate material.

## 3    Results

### 3.1    Isotopic and elemental tracers

### 3.1.1    Primary producers, sedimentary and suspended particulate organic matter

Mangrove leaves showed the most depleted $\delta^{13}C$ signatures and the highest C:N ratios of all organic matter sources in both transects and seasons (Fig. 2). Macroalgae, followed by seagrasses, presented higher $\delta^{13}C$ than mangroves, and the values increased progressively moving from the mangroves towards the sea. In addition, both macroalgae and seagrasses showed the most depleted $^{13}C$, as well as a wider $\delta^{13}C$ range, in transect B compared with transect A (Fig. 2). Isotopic and elemental zooplankton values were more homogeneous at both spatial and temporal scales.

$\delta^{13}C$ of sedimentary organic matter ($\delta^{13}C_{SOM}$) increased drastically along the land-to-sea transects (Fig. 3-a). $\delta^{13}C_{SOM}$ of the landward stations (M, IA) was lower overall in transect B than in A, while it was more homogenous in the other stations and showed a marked increase in the coral reefs (CR). Sedimentary C:N ratio (C:N$_{SOM}$) also showed a marked spatial pattern in the bay, decreasing steadily from mangroves (M) to coral reefs (CR) (Fig. 3-b). A striking seasonal difference was observed in the mangrove station of transect A, where the lowest value (10.2) was recorded in the rainy season and the highest (21.5) in the



dry. Unlike SOM, $\delta^{13}C_{SPOM}$ and C:N$_{SPOM}$ were overall more homogeneous among stations (Fig. 3-c,d) except for the highest values recorded in the samples collected in seagrass beds (SB).

### 3.1.2 Mixing models

Bayesian mixing model outcomes provided the ranges of possible contributions (95th percentile intervals) of organic matter sources to sedimentary (SOM) and suspended particulate (SPOM) organic matter. Although the percentile intervals were wide and often overlapped, indicating a potential contribution of multiple sources and a non-negligible level of uncertainty in some cases, overall the contribution of mangroves to SOM was high in the whole bay up to the seagrass beds, while it decreased sharply in coral reefs (Fig. 4). In the dry season, the mangrove contribution to SOM was higher in the mangrove station (M), especially in transect B (10.8-74.4%), while in the rainy season it was higher in mangrove, intertidal and seagrass bed stations along transect A and in the former two in transect B. The contribution of macroalgae and seagrasses to SOM was generally comparable except in coral reefs, where their influence was more important overall in both transects.

The contribution of mangrove-derived organic matter to SPOM was important in Gazi Bay up to the coral reefs (Fig. 5), without any relevant seasonal pattern. Along transect A, the mangrove contribution was higher than that of the other sources within the bay (15.1-74.5%) and then diminished in CR, becoming comparable to the other sources. Along transect B, mangrove contribution was fairly important in all stations, with a peak in IA in both seasons (13.9-83.8%).

### 3.2 Fatty acids

### 3.2.1 Primary producers

The different species of primary producers within each group (mangroves, seagrasses, brown and red macroalgae) showed no significant differences in their FA profiles between seasons, transects, and stations (PERMANOVA p > 0.05). Thus, the data of each species were further elaborated as the above groups (Tab. 1). Saturated fatty acids (SFAs) were particularly abundant in mangroves and brown and red macroalgae, due to the contribution of 16:00 and 14:00 FAs. Long chain fatty acids (LCFAs), typical biomarkers of mangroves (Kelly and Scheibling, 2012; Meziane et al., 2006), were particularly abundant



in mangrove leaves. Polyunsaturated fatty acids (PUFAs) consisted almost exclusively of 18:2 n6 and 18:3 n3 FAs in seagrasses and mangroves, and 20:5 n3 in red macroalgae, while 18:4 n3, 20:4 n6 and 20:2 n6 were also abundant in brown macroalgae, consistent with the literature (Kelly and Scheibling 2012 and references therein). Among monounsaturated fatty acids (MUFAs), overall 18:1 n9, followed by 16:1 n7, were the most abundant in all primary producers.

Primary producer groups were well separated in the multivariate ordination (Fig. 6), with mangroves and seagrasses in the left part of the plot, and brown and red macroalgae in the right part. Indeed, there were significant differences among primary producer groups based on their FA profile (PERMANOVA: df = 3; MS: 15076; p < 0.001). All pairwise comparisons were also highly significant (p < 0.001). SIMPER analysis revealed that mangroves differed from the other primary producers in having a higher contribution of LCFAs (Tab. 2). Seagrasses and mangroves were distinguished by 18:3 n3 and 18:2 n6, which contributed more to the seagrass than to the mangrove within-group similarity. 20:5 n3 was abundant in red macroalgae, 18:4 n3, 20:4 n6 and 20:2 n6 in brown macroalgae.

## 3.2.2 Sedimentary and suspended particulate organic matter

Forty and thirty-five individual fatty acids were identified respectively in sedimentary and suspended particulate organic matter. They included i) straight- and branched-chain saturated FAs (SFAs) dominated by 16:00, 18:00 and long chain fatty acids (LCFAs), ii) monounsaturated fatty acids (MUFAs), mainly constituted by 16:1 n7, 18:1 n7 and 18:1 n9, iii) polyunsaturated fatty acids (PUFAs) with a high contribution of 18:2 n6, 20:5 n3 and 22:4 n6, iv) cyclopropyl (CY) FAs, and v) hydroxyl (OH) FAs (see supplementary material).

The contribution of LCFAs to SOM was markedly high in the mangrove stands of transect B, comprising up to 40% of the total FAs, and decreased toward coral reefs (Fig. 7). In contrast, in transect A, LCFAs were between approximately 5 and 10% in the mangrove stand and increased in the adjacent habitats (IA, SB), especially in the rainy season. Relative contributions of seagrass (18:2 n6 and 18:3 n3), brown (18:1 n9 and 18:4 n3) and red macroalgae (20:5 n3) biomarkers were fairly similar in the whole bay. The relative contribution of bacterial fatty acids, BaFAs (branched 15:0 and 17:0, and 18:1 n7; Kelly and Scheibling



2012 and reference therein), was notable and slightly greater in IA and SB than in mangrove stands and coral reefs, peaking in transect B.

FA profiles of suspended particulate organic matter were also characterised by a large contribution of LCFAs and BaFAs (Fig. 8). In the dry season, they showed a similar spatial trend in both transects, with increasing proportions from mangroves to seagrass beds, followed by a decrease in coral reefs. In contrast, in the rainy season these FAs decreased seaward. Mean contribution of brown and red macroalgae- and seagrass-derived organic matter to suspended particulate FAs was fairly homogenous in Gazi Bay in both seasons. The contribution of 20:1 and 22:1, biomarkers of zooplankton (Alfaro et al., 2006; Bachok et al., 2003), was also fairly uniform in the whole bay.

## 4 Discussion

### 4.1 Organic matter sources

Elemental and isotopic signatures of mangrove leaves were consistent with documented data from Gazi Bay (Hemminga et al., 1994; Rao et al., 1994) and other mangrove forests (Kristensen et al. 2008). The high C:N ratio of leaves (overall, between 100 and 200) is attributable to their senescent status, due to resorption of nitrogen by the mangrove tree during leaf senescence (Woitchik et al., 1997). After the leaf drops onto the soil, consistent nitrogen enrichment *versus* carbon occurs during decomposition, as a result of a number of physical and chemical transformations (litter breakdown by benthic organisms and microbial remineralization; Bosire et al., 2005). These processes explain why the C:N ratio of mangrove sediments was one order of magnitude lower than that of recently fallen senescent leaves, consistent with the literature (Davis et al., 2003; Dunn et al., 2008; Woitchik et al., 1997).

Mangrove leaves showed low carbon isotopic signatures characteristic of $C_3$ vegetation (Bouillon et al., 2008), while both macroalgae and seagrasses were $^{13}C$ enriched compared to mangroves, and variable along the land-to-sea transects. Indeed, marked gradients were observed, with the most depleted values in the landward station close to mangrove stands, especially in transect B, and the most enriched ones in the seaward station. A similar enrichment was already observed in Gazi Bay and other tropical areas (Hemminga et al., 1994; Lugendo et al., 2007) as well as parallel changes in $\delta^{13}C_{DIC}$ (Alongi, 2014; Maher et al., 2013). $\delta^{13}C_{DIC}$ is typically more negative close to mangroves as a result of the intense localized



mineralization of mangrove detritus (Bouillon et al., 2007) and increases seaward due to the increased contribution of oceanic DIC, whose $\delta^{13}$C is typically around 0‰ (Bouillon et al., 2008). The among-creek variability, with both macroalgae and seagrasses noticeably more $^{13}$C depleted in Kinondo Creek than in Kidogoweni, may be due to many factors, but the different settings of the two creeks must account for most of this variability. In particular, the lack of fresh water in Kinondo, coupled with its lower water volume, suggests that terrestrial and mangrove-derived DIC export is less effective when driven by tidal pumping only. The higher salinity (Kitheka et al., 1996) and lower $\delta^{13}$C$_{DIC}$ in Kinondo waters (Bouillon et al., 2007) support this hypothesis.

## 4.2   Export of mangrove organic matter

Mangrove-derived organic matter contributed greatly to the sedimentary pool at Kinondo Creek and the adjacent intertidal area, as revealed by isotope mixing models and the great abundance of long chain fatty acids (LCFAs). Moving seaward, mangrove contribution progressively decreased due to efficient suspended particle trapping by the seagrass canopy (Hemminga and Duarte, 2008), and dropped steeply in the coral reef, consistent with Hemminga et al. (1994). This pattern suggests that mangrove stands at the Kinondo mouth acted as a retention site where mangrove organic carbon tends to accumulate into sediments. In contrast, freshwater discharge from the Kidogoweni Creek seemed to enhance the export of mangrove material, contributing up to 50% to the sediment pool in the seagrass area, as revealed by mixing models and confirmed by the higher relative abundance of LCFAs in both stations within the bay. A certain influence of the Mkurumuji River to the organic matter pool of the seagrass beds cannot be excluded. Indeed, the Mkurumuji River is characterised by high water flows in the south-western part of the bay peaking at 16.7 m$^3$ s$^{-1}$ during the rainy season (Kitheka et al., 1996) and a large catchment area covered by dense C$_4$ vegetation and fewer mangrove trees (Bouillon et al., 2007). The highest $\delta^{13}$C and C:N values of SPOM found in the seagrass beds in the rainy season corroborate this assumption, since C$_4$ plants have higher $\delta^{13}$C and C:N ratio than mangroves (Lamb et al., 2006).

The among-site variability observed in Gazi Bay is consistent with the pattern recently described by Adame and Lovelock (2011): mangroves dominated by tidal pumping, like the Kinondo stand, are characterised by bidirectional flows that favour retention of autochthonous material and tight interlinkage



between mangroves and adjacent seagrass beds. Higher retention of autochthonous organic matter within a mangrove forest has also been reported in arid mangrove systems in the Arabian Gulf with almost no freshwater input (Walton et al., 2014). On the other hand, riverine mangroves, like the Kidogoweni stand, are dominated by unidirectional flows that favour outwelling (Adame and Lovelock, 2011).

The different dominant vegetation in Kidogoweni and Kinondo creeks (*Avicennia marina vs. Rhizophora mucronata;* Neukermans et al. 2008) may also affect the magnitude of organic matter export from mangrove forest. Higher export from forests dominated by *Avicennia* sp., compared to those dominated by *Rhizophora* sp. has been observed in Brazilian forests and related to different detritus decomposition rates (Lacerda et al., 1995). This is plausible also because the sediment C:N ratio in the Kinondo stand

was constant across seasons, while in the Kidogoweni stand it was almost twofold higher in the dry season than in the rainy ($20.8 \pm 0.6$ *vs.* $11.6 \pm 1.3$). Given that $\delta^{13}C$ was constant across time, we may infer a greater contribution of decomposed mangrove litterfall to the sediment in the rainy season. Indeed, decomposition processes are known to reduce the C:N ratio, but not $\delta^{13}C$ (Bosire et al., 2005; Dunn et al., 2008), and are enhanced by a greater supply of nutrients through freshwater flooding during rainfall

(Woitchik et al., 1997).

Contrarily to sediments, mangrove-derived organic matter was recorded in suspended particulate organic matter throughout the whole bay up to the coral reefs, as indicated by the outcomes of mixing models, C:N ratio and LCFA relative abundances. This contrasts with previous findings that indicated a limited contribution of mangrove material to the SPOM pool in the southern part of Gazi Bay (Bouillon et al.,

2007; Hemminga et al., 1994) and may be due to the timing of sampling, which was carried out mostly during ebb tide. Strong downstream ebb currents have been shown to enhance mangrove detritus export and the displacement of mangrove and seagrass waters toward coral reefs (Kitheka, 1997). Despite the buffering role of seagrass beds in preventing the direct link between mangrove and oceanic waters, we infer that high mangrove export coupled with a high rate of water exchange at spring ebb tide did not

impede suspended mangrove material reaching the coral reef inner area.

**4.3 Contribution of other primary producers to the sediment and suspended particulate pool**





Mixing model results revealed a widespread contribution of seagrasses and macroalgae to both sediment and particulate organic matter pools. In particular, the macroalgae contribution was slightly higher than that of seagrasses within the bay and similar or lower in the coral reef. Fatty acids confirmed the spread of seagrass and macroalgae-derived organic matter across Gazi Bay.

With their wide distribution and high productivity as well as their ecological function for marine fauna (e.g. Heck et al. 2008, Gillis et al. 2014), the importance of seagrasses in providing marked carbon input to adjacent and even distant habitats has been widely described. However, relatively few data are available on the productivity and contribution of macroalgae to the carbon budget. Moreover, macroalgae are seldom end-members in mixing models, despite studies on the important trophic and budgeting role of

macroalgae associated with mangroves (Koch and Madden, 2001), seagrasses (Heck et al., 2008) and coral reefs (Wernberg et al., 2006). A large quantity of drifting macroalgae, especially *Sargassum* sp., is transported passively by tides in Gazi Bay (pers. obs.; Coppejans et al. 1992) and may account for the large and widespread contribution of brown macroalgae to sediment and, in particular, to the suspended particulate organic matter pool, as revealed by detection-specific fatty acid biomarkers (18:1n9 + 18:4n3).

FA ratios, 16:1n-7/16:0 and 20:5n-3/22:6n-3, markers of dinoflagellate/diatom organic matter (Alfaro et al., 2006), were lower than established thresholds (1.6 and 1 respectively; Alfaro et al. 2006) in all samples, suggesting dinoflagellate dominance. Although local production of pelagic and benthic microalgae may be an important part of sediment and suspended particulate pools, the low relative abundance of 22:6n3 (SOM: $0.9 \pm 0.6\%$; SPOM: $1.3 \pm 0.8\%$) indicated a negligible role.

In contrast, the relative abundance of bacterial biomarkers (branched FAs and 18:1n7) was particularly important in both sediment and suspended particulate pools, indicating a central role for bacteria as mineralizers of organic detritus in Gazi Bay. Cyclopropyl (CY) and hydroxyl (OH) fatty acids have been ascribed to gram-negative bacteria (Kaur et al., 2005) and were also found in both compartments. The abundance of bacteria is not surprising, as they are able to process most of the energy and nutrients in

tropical systems (Alongi, 1994). The higher percentage of bacterial biomarkers within the bay suggests a greater benthic mineralization in the intertidal and seagrass bed sediments. This is consistent with a previous study carried out in Gazi Bay using $\delta^{13}C$ analysis of bacteria-specific PLFAs (phospholipid fatty acids), which found that mangrove carbon was a significant source for bacteria mineralization in the




adjacent seagrass beds (Bouillon et al. 2004). The high percentage of bacterial biomarkers in the intertidal area at the Kinondo creek mouth suggests greater benthic mineralization, confirming the higher trapping ability of this mangrove stand and tight coupling with the adjacent area.

**5. Conclusion**

Using FA profiles, elemental and SI signatures, this study showed that small-scale variability in the environmental settings of the mangrove forest has a crucial role in influencing the export of mangrove-derived organic matter to adjacent habitats. In Gazi Bay the short and shallow Kinondo Creek lacks significant freshwater input and, being influenced by tidal pumping only, acted as a retention site where mangrove material accumulated in the forest sediment and the contiguous intertidal area. In contrast, the longer and deeper Kidogoweni Creek, receiving high freshwater input especially in the rainy season, contributed to the export of mangrove material to seagrass beds within the bay. Suspended particulate organic matter revealed a greater contribution of mangrove material up to the coral reefs during ebb tide, with almost no influence from the creek settings. Instead, the riverine efflux of the Mkurumuji River, which flows into the southern side of the bay, seemed to contribute more to the suspended particulate pool with $C_4$ vegetal material. In addition, a notable importance of macroalgae-derived material across the bay was detected. In particular, brown macroalgae seemed to be the most important, probably due to the high biomass of drifting *Sargassum binderi*. Fatty acids were particularly well suited to detecting the great bacterial biomass in both the sedimentary and suspended particulate organic matter, suggesting a central role for bacteria as mineralizers of organic detritus in Gazi Bay. These findings confirm the importance of taking into account geomorphological settings and seasonal variability when addressing the question of organic matter export from mangroves in tropical systems. Given the major influence that organic matter exchange across ecosystem boundaries has on organic matter availability and consumption by fauna in trophic webs, future research should be focused on understanding how this small-scale spatial variability may affect trophic dynamics. Further, future research needs to investigate changes in marine productivity with changes in land use activities within and adjacent to Gazi bay, including mining, seaweed farming, and large-scale sugar plantation.





## Acknowledgement

This work was financially supported by the Research Grant "Luigi e Francesca Brusarosco 2008" (Luigi e Francesca Brusarosco Association – Research centre Space, University of Milano – Italian Association of Ecology S.It.E) and by the Flagship Project "RITMARE – Italian Research for the Sea" funded by the Italian Ministry of Education, University and Research. Special thanks to Dr J. Bosire for providing facilities and logistical support during the study. We are also grateful to the people of Gazi who assisted in the fields and to the staff of KMFRI for support during laboratory activities.

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

**Tables**

Table 1. Mean (± sd) relative abundance of fatty acids (% of total FAs) in primary producers. FAs <0.5% in all the samples are omitted.

| Primary producer group | Mangrove | | Seagrass | | | | Brown macroalgae | | | Red macroalgae | |
|---|---|---|---|---|---|---|---|---|---|---|---|
| Species | R. mucronata | S. alba | C. rotundata | C. serrulata | E. acoroides | T. ciliatum | T. conoides | D. cervicornis | S. binderi | H. cornuta | G. salicornia |
| Fatty acid | n= 8 | n= 6 | n= 5 | n= 13 | n= 10 | n= 14 | n= 4 | n= 4 | n= 4 | n= 6 | n= 4 |
| 14:00 | 8.0±1.7 | 2.9±0.5 | 1.1±0.3 | 1.0±0.6 | 0.7±0.3 | 0.6±0.4 | 7.5±0.6 | 13.3±6.0 | 11.6±0.6 | 13.2±3.0 | 1.9±0.3 |
| 15:00 | 0.4±0.1 | 0.3±0.1 | 0.7±0.6 | 0.5±0.3 | 0.4±0.3 | 0.2±0.1 | 0.6±0.1 | 0.9±0.2 | 0.6±0.0 | 0.6±0.2 | 0.2±0.2 |
| i-16:00 | 0.7±0.3 | 0.7±0.6 | 0.4±0.0 | 0.3±0.0 | 0.4±0.1 | 0.3±0.1 | 0.0±0.0 | 0.3±0.3 | 0.0±0.0 | 0.1±0.1 | 0.2±0.0 |
| 16:00 | 44.3±3.0 | 30.0±5.3 | 21.5±3.8 | 19.2±3.5 | 20.2±3.1 | 16.2±2.6 | 41.6±1.9 | 26.4±1.8 | 36.4±2.4 | 53.9±3.4 | 68.0±2.5 |
| i-17:00 | 0.0±0.0 | 0.0±0.0 | 0.0±0.0 | 0.0±0.0 | 0.0±0.0 | 0.0±0.0 | 0.6±0.1 | 0.3±0.4 | 0.6±0.1 | 0.1±0.2 | 0.0±0.0 |
| 17:00 | 1.7±0.4 | 0.9±0.2 | 0.6±0.5 | 0.8±0.1 | 0.2±0.3 | 0.4±0.1 | 0.0±0.0 | 0.3±0.2 | 0.0±0.0 | 0.1±0.2 | 0.0±0.0 |
| 18:00 | 5.2±0.5 | 4.0±0.7 | 4.6±1.3 | 2.1±0.7 | 4.4±1.2 | 1.7±0.4 | 1.3±0.3 | 1.6±0.1 | 0.8±0.1 | 1.2±0.3 | 1.3±0.1 |
| 20:00 | 1.0±0.1 | 4.3±2.8 | 0.0±0.0 | 0.1±0.1 | 0.5±0.2 | 0.4±0.1 | 0.3±0.1 | 0.7±0.2 | 0.3±0.0 | 0.0±0.0 | 0.0±0.0 |
| LCFAs (> 22:00) | 7.0±1.2 | 9.3±1.0 | 2.0±1.1 | 1.9±0.7 | 1.3±0.5 | 1.6±0.5 | 1.2±0.4 | 0.5±0.6 | 0.4±0.1 | 0.0±0.0 | 0.0±0.0 |
| ΣSFAs | 68.5±4.3 | 52.5±3.8 | 31.0±3.4 | 26.1±3.7 | 28.3±3.0 | 21.5±2.8 | 53.4±1.8 | 44.5±4.1 | 51.1±2.3 | 69.6±2.4 | 71.7±2.0 |
| | | | | | | | | | | | |
| 16:1 n7 | 0.5±0.2 | 0.2±0.2 | 2.9±1.1 | 4.2±1.0 | 2.4±1.3 | 3.6±0.8 | 1.9±0.1 | 4.2±2.6 | 5.1±1.1 | 1.2±0.6 | 0.8±0.2 |
| 18:1 n9c | 7.7±1.4 | 8.6±3.2 | 5.4±1.5 | 4.8±2.9 | 1.1±0.9 | 2.3±0.8 | 12.6±0.9 | 15.4±1.0 | 10.2±0.6 | 5.4±0.4 | 3.9±0.4 |
| 18:1n7 | 0.5±0.4 | 0.7±0.3 | 0.9±1.0 | 0.5±0.3 | 0.7±0.5 | 0.8±0.6 | 0.1±0.1 | 0.6±0.4 | 0.1±0.0 | 0.7±0.2 | 0.6±0.3 |
| 20:1 n9 | 0.0±0.0 | 0.0±0.0 | 0.1±0.1 | 0.1±0.1 | 0.2±0.1 | 0.2±0.2 | 0.5±0.1 | 0.1±0.1 | 0.0±0.0 | 0.0±0.0 | 0.0±0.0 |
| 22:1 n9 | 0.0±0.0 | 0.0±0.0 | 0.0±0.1 | 0.1±0.1 | 0.1±0.1 | 0.1±0.1 | 0.2±0.1 | 0.2±0.1 | 0.2±0.2 | 0.1±0.1 | 0.5±0.5 |
| 24:1 n9 | 0.0±0.0 | 0.0±0.0 | 0.0±0.1 | 0.1±0.1 | 0.1±0.1 | 0.1±0.1 | 0.3±0.2 | 0.3±0.1 | 0.2±0.3 | 0.2±0.3 | 0.1±0.1 |
| ΣMUFAs | 8.6±1.6 | 9.5±3.5 | 9.3±1.7 | 9.9±2.3 | 4.6±0.8 | 7.1±1.7 | 15.6±0.9 | 20.8±4.1 | 15.8±0.6 | 7.5±0.8 | 5.9±0.7 |
| | | | | | | | | | | | |
| 18:2 n6 | 6.3±2.5 | 15.2±6.5 | 20.1±4.6 | 23.9±4.2 | 24.2±7.9 | 27.1±6.2 | 2.3±0.1 | 1.7±0.7 | 2.9±0.7 | 0.9±0.3 | 0.3±0.1 |
| 18:3 n3 | 15.7±3.3 | 21.9±6.3 | 37.9±0.5 | 38.6±4.3 | 41.0±5.6 | 42.5±5.0 | 1.2±0.4 | 1.3±0.2 | 1.3±0.2 | 0.3±0.1 | 0.6±0.2 |
| 18:3 n6 | 0.0±0.0 | 0.0±0.0 | 0.0±0.0 | 0.0±0.0 | 0.0±0.0 | 0.0±0.0 | 2.5±0.1 | 2.8±0.1 | 2.5±0.0 | 1.4±0.4 | 1.1±0.0 |
| 18:4 n3 | 0.0±0.0 | 0.0±0.0 | 0.0±0.0 | 0.0±0.0 | 0.0±0.0 | 0.0±0.0 | 10.4±0.2 | 14.1±4.8 | 12.0±0.4 | 0.0±0.0 | 0.0±0.0 |
| 20:2 n6 | 0.1±0.1 | 0.1±0.1 | 0.2±0.4 | 0.1±0.1 | 0.2±0.1 | 0.2±0.2 | 5.3±0.1 | 5.3±0.3 | 5.2±0.0 | 1.5±0.4 | 1.6±0.6 |
| 20:3 n6 | 0.0±0.0 | 0.0±0.0 | 0.1±0.1 | 0.1±0.1 | 0.1±0.1 | 0.1±0.1 | 0.3±0.1 | 0.5±0.1 | 0.3±0.0 | 0.2±0.1 | 0.8±0.2 |
| 20:4 n6 | 0.0±0.0 | 0.0±0.0 | 0.0±0.0 | 0.1±0.1 | 0.1±0.1 | 0.1±0.1 | 7.1±0.7 | 7.3±3.8 | 6.6±1.2 | 3.0±1.0 | 0.0±0.0 |
| 20:5 n3 | 0.0±0.0 | 0.0±0.0 | 0.0±0.0 | 0.1±0.2 | 0.1±0.1 | 0.1±0.1 | 1.0±0.1 | 0.3±0.3 | 1.6±0.2 | 14.6±1.4 | 17.0±1.8 |
| 22:4 n6 | 0.5±0.1 | 0.5±0.3 | 0.3±0.1 | 0.3±0.1 | 0.3±0.1 | 0.3±0.1 | 0.0±0.0 | 0.4±0.1 | 0.0±0.0 | 0.0±0.0 | 0.0±0.0 |
| ΣPUFAs | 22.7±5.0 | 37.8±2.4 | 58.9±4.6 | 63.4±4.5 | 66.4±3.5 | 70.7±2.9 | 30.4±1.0 | 34.2±1.5 | 32.6±1.9 | 21.9±1.9 | 21.6±2.5 |



Table 2. Results of similarity percentage analysis (SIMPER) of FA profiles of primary producer groups. Data were not transformed prior to analysis.

| Primary producer group | FAs | Average abundance (%) | Contribution to similarity (%) |
|---|---|---|---|
| **Mangroves** | 16:00 | 38.2 | 40.8 |
| average similarity: 81.7% | 18:3 n3 | 18.4 | 18.8 |
| | LCFAs (>22) | 8.0 | 8.7 |
| | 18:1 n9 | 8.1 | 8.2 |
| | 18:2 n6 | 10.1 | 8.0 |
| **Seagrasses** | 18:3 n3 | 40.4 | 43.6 |
| average similarity: 86.6% | 18:2 n6 | 24.6 | 24.3 |
| | 16:00 | 18.7 | 19.2 |
| **Brown macroalgae** | 16:00 | 34.8 | 36.0 |
| average similarity: 85.5% | 18:1 n9 | 12.7 | 13.2 |
| | 18:4 n3 | 12.2 | 12.4 |
| | 14:00 | 10.8 | 10.1 |
| | 20:4 n6 | 7.0 | 6.7 |
| | 20:2 n6 | 5.3 | 6.1 |
| **Red macroalgae** | 16:00 | 59.6 | 63.5 |
| average similarity: 86.4% | 20:5 n3 | 15.5 | 16.7 |
| | 14:00 | 8.7 | 5.8 |





**Figures**

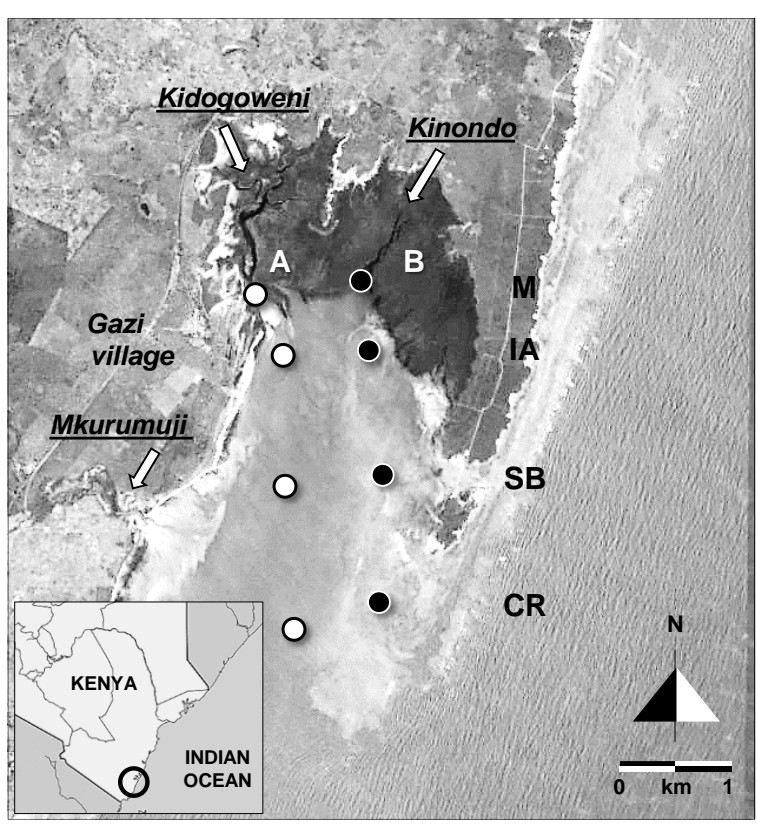

Figure 1. Gazi Bay. Sampling stations lying along two land-to-sea transects are shown: mangrove forest (M), intertidal area (IA), seagrass bed (SB), coral reef (CR). Transect A (white circles) is in correspondence with Kidogoweni Creek; transect B (black circles) in correspondence with Kinondo Creek.





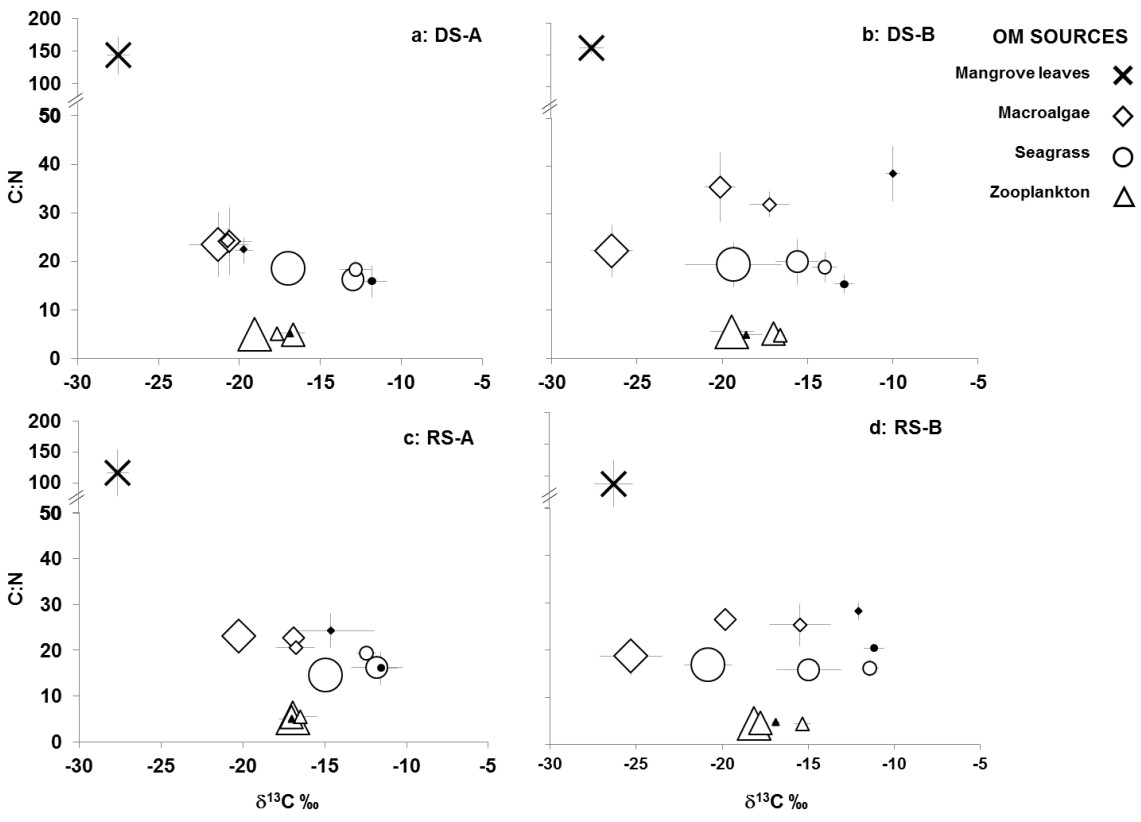

Figure 2. C:N ratio *versus* $\delta^{13}C$ (‰) of organic matter sources from the four sampling stations (symbols with decreasing size from mangroves to intertidal areas, seagrass beds and coral reefs along transects A and B in dry (DS) and rainy seasons (RS). Mangroves (cross), macroalgae (diamond), seagrasses (circle) and zooplankton (triangle).




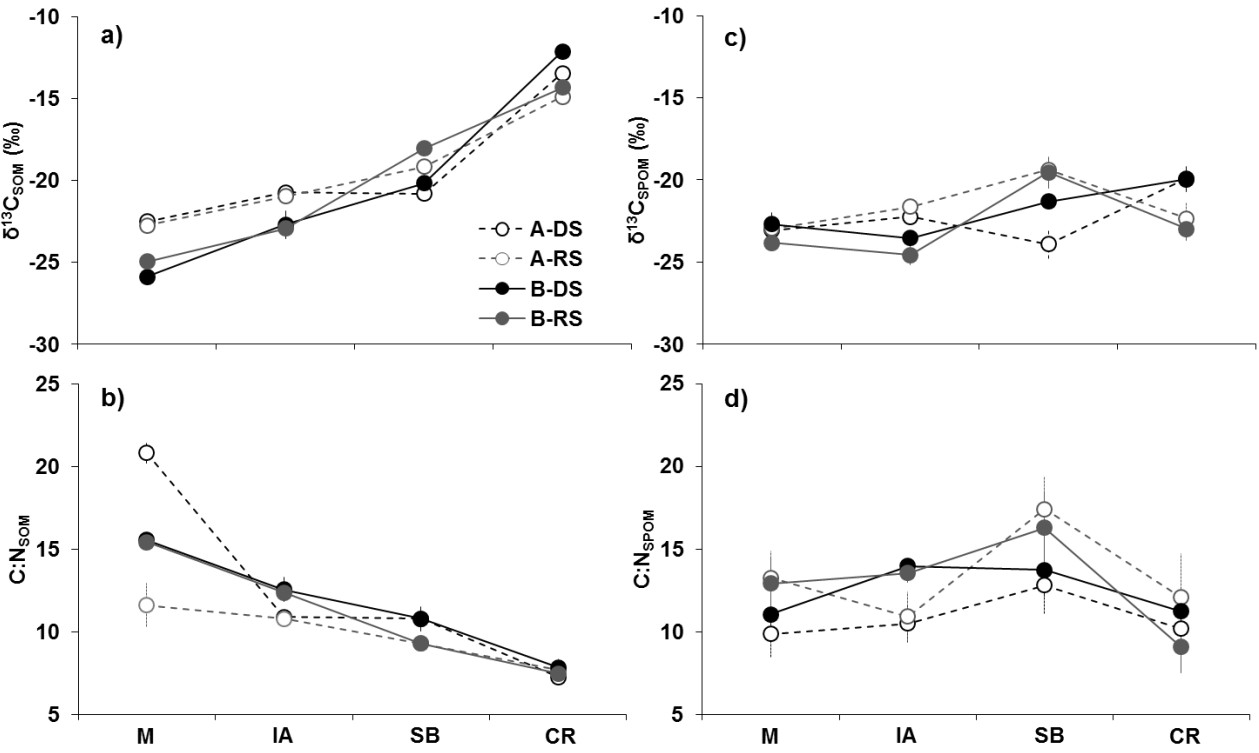

Figure 3. $\delta^{13}C$ (‰) and C:N ratio of sedimentary organic matter (SOM; a, b) and suspended particulate organic matter (SPOM; c, d) from the four sampling stations (mangroves M, intertidal areas IA, seagrass beds SB, coral reefs CR). Transects A and B are indicated with open and filled circles respectively. Dry (DS) and rainy seasons (RS) are indicated in black and grey circles respectively.



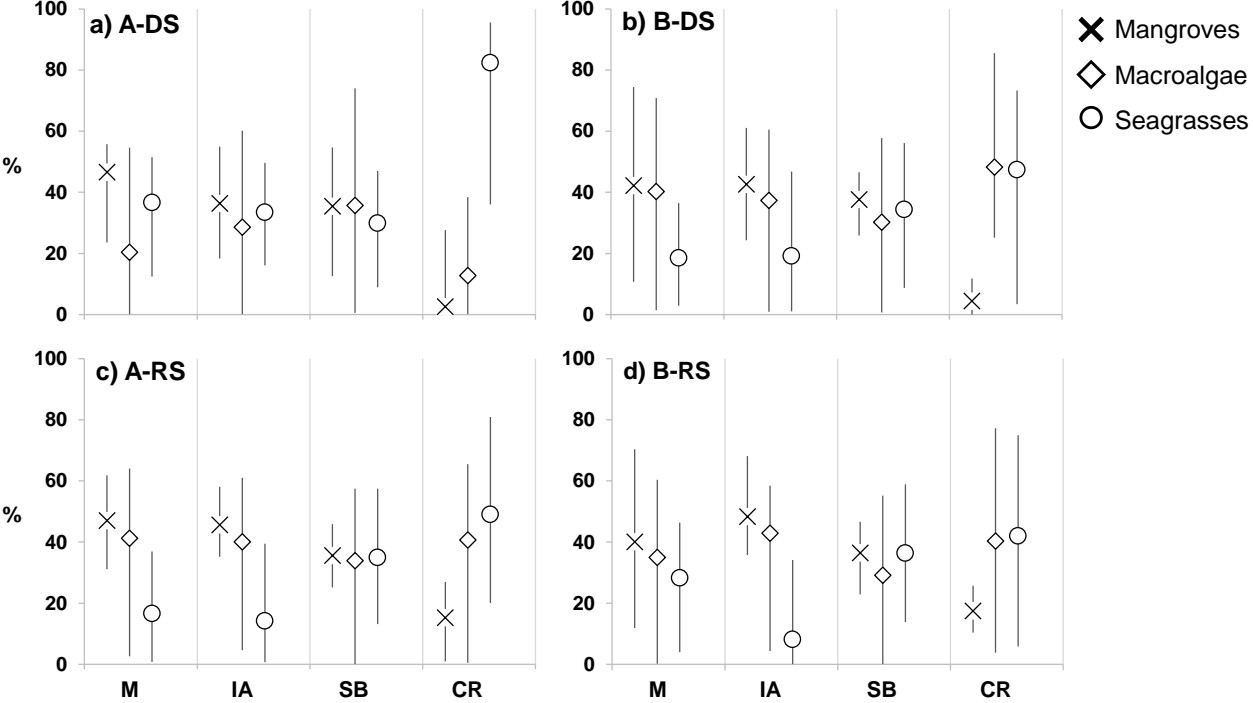

Figure 4. Organic matter source contribution (mode ± 95$^{th}$ credibility interval) to SOM in the four sampling stations (mangroves M; intertidal areas IA; seagrass beds SB; coral reefs CR) along transects A and B in dry (DS) and rainy season (RS). Mangroves (cross), macroalgae (diamond) and seagrasses (circle).





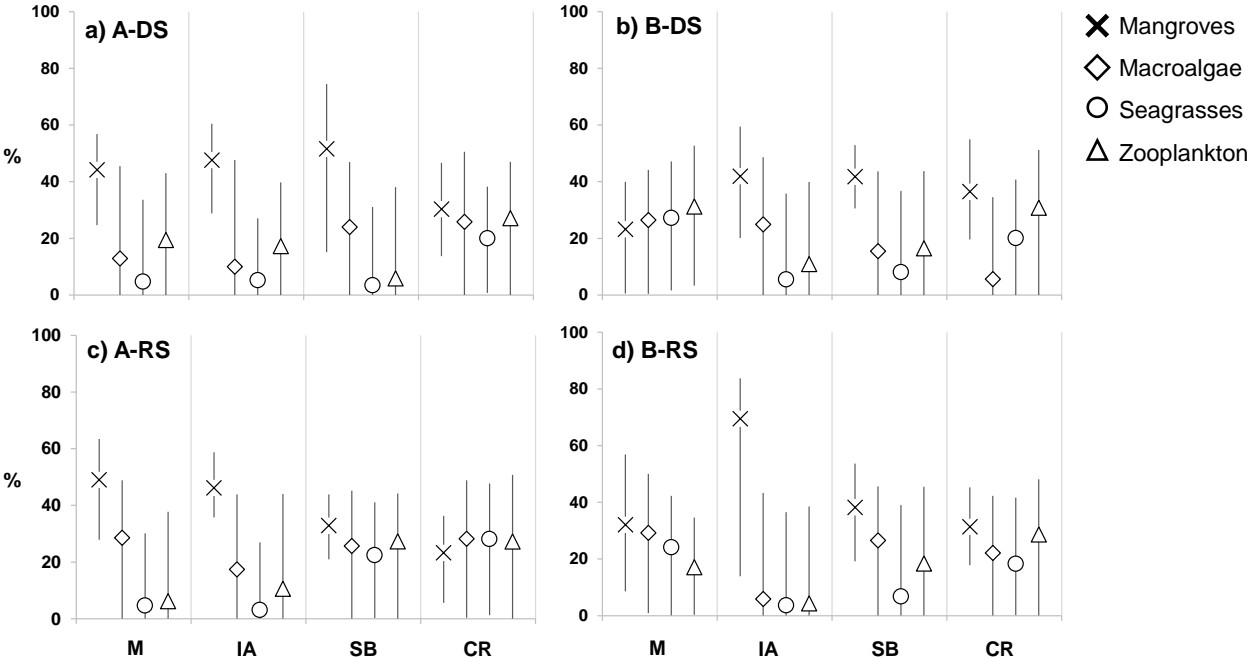

Figure 5. Organic matter source contribution (mode ± 95$^{th}$ credibility interval) to SPOM in the four sampling stations (mangroves M; intertidal areas IA; seagrass beds SB; coral reefs CR) along transects A and B in dry (DS) and rainy season (RS). Mangroves (cross), macroalgae (diamond), seagrasses (circle) and zooplankton (triangle).



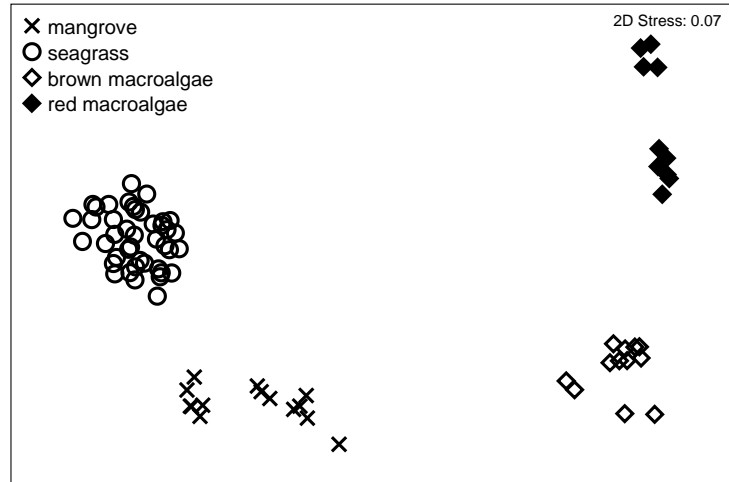

Figure 6. nMDS based on FA profiles of primary producer groups: mangroves (cross); brown macroalgae (white diamond); red macroalgae (black diamond); seagrasses (circle).





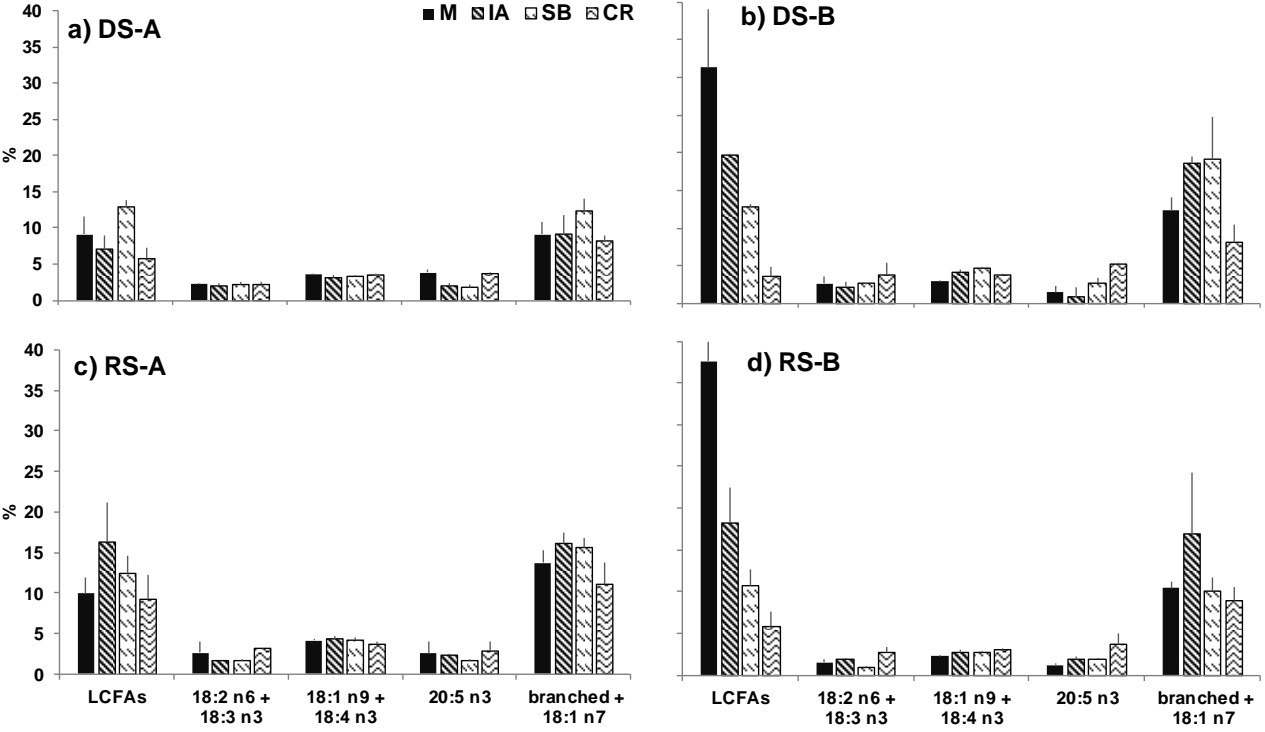

Figure 7. Mean (± s.d.) relative abundance of biomarker FAs (% of total FAs) in SOM. Selected biomarkers include: LCFAs for mangroves, 18:2 n6 + 18:3 n3 for seagrasses, 18:1 n9 + 18:4 n3 for brown macroalgae, 20:5 n3 for red macroalgae and branched + 18:1 n7 for bacteria.



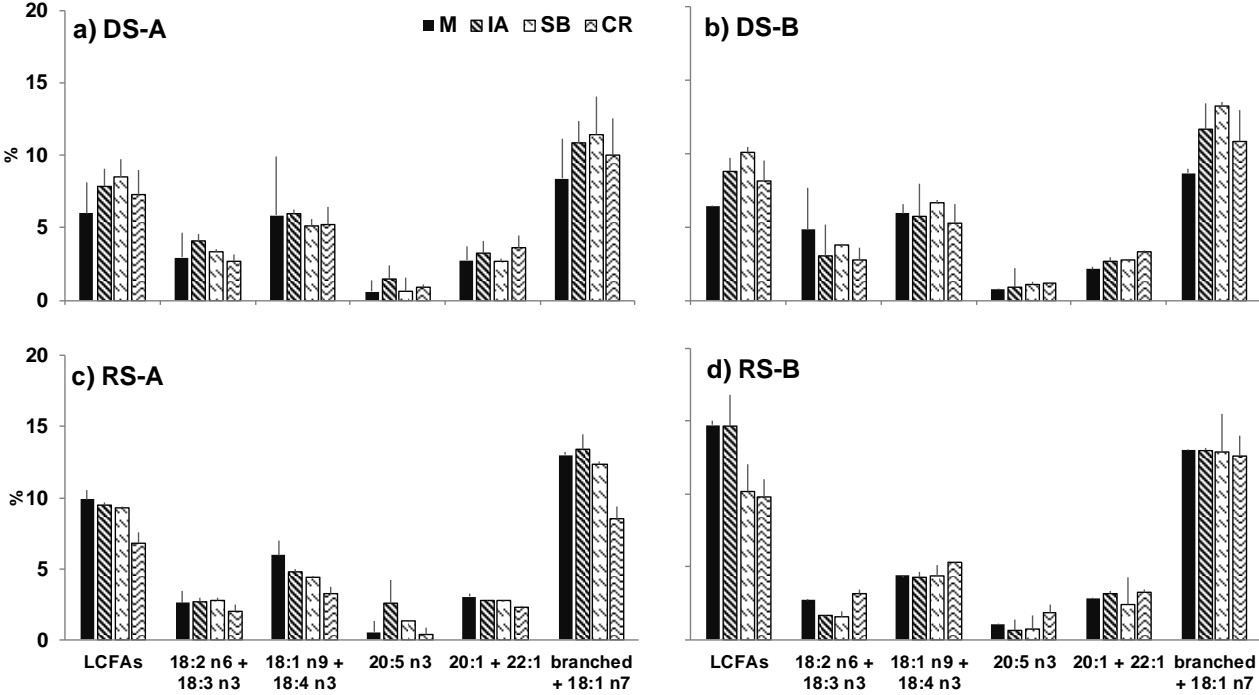

Figure 8. Mean (± s.d.) relative abundance of biomarker FAs (% of total FA) in SPOM. Selected
biomarkers include: LCFAs for mangroves, 18:2 n6 + 18:3 n3 for seagrasses, 18:1 n9 + 18:4 n3 for brown
macroalgae, 20:5 n3 for red macroalgae and branched + 18:1 n7 for bacteria.