# Peer review of "Small-scale variability of geomorphological settings influences mangrove-derived organic matter export in a tropical bay"

_Biogeosciences, 2016_

## Referee Comment (RC1) · Anonymous Referee #1 · 22 Sep 2016

Comments

The authors of this paper have investigated changes in mangrove outwelling to seagrass beds and coral reefs over seasons in Gazi Bay, Kenya. Mangrove forests are highly productive-ecosystems and have important implications for the exchange of organic material in the tropical coastal seascape. They have combined more traditional analyses (isotopes) with modern techniques (fatty acids) to explain the source contributions and spatial distribution of organic material across the bay. In addition they also investigated how physical factors such as tidal exchanges, river runoff and season may affect the exchange of organic material between different habitats.

The authors state they have found significant differences in transport of organic material from mangrove forest creeks to adjacent systems. This was primarily due to river runoff and tidal transport. However the river runoff creek was influenced strongly by the rainy season. Interestingly the authors found that macroalgae also had a contribution to organic material, this ecosystem is often ignored in the tropical seascape.

The paper is of an interesting subject with new techniques applied. There is a dearth of data regarding the outwelling of mangroves with physical factors taken into account especially with respect to connectivity. It is quite well written and structured. The abstract and introduction are well done and flow nicely. There are minor comments on these sections. I admire the authors for doing such a through job on this study, they have tried to expand our understanding using new techniques in addition they must have worked very hard. It is not easy to complete fieldwork over two seasons and fatty acids are very labor intensive. However, I have some concerns regarding the methodology, results and major concerns (section 4.2) regarding the discussion.

Methodology

No statistics were used to compare the isotopes sources. This makes it difficult to understand the patterns in the data the authors state. The changes the authors saw in the organic material sources did not seem statistically significant nor could was it shown in the figures. A statistical test would resolve this.

Results

The figures depicting the results are numerous and difficult to understand. Several of the tables/figures could go into the supplementary section (even though there is already data in this section), other figures need to be edited for clarity and others could be drastically improved by a different method of illustrating them. Several issues in the discussion may be due to the difficulty in understanding the figures. In addition I and I think others would find it easier if at some points the authors specified which creek applied to transects A and B, it was annoying to keep on referring to the methodology to understand which was which.

For example, section 3.1.2, line 7- I am not sure if "often" is the correct word to use. They seemed to be almost always overlapping.

Table 1 and 2 could go in the supplementary material. Table 1 especially should be in the supplementary material.

Figure 1 is badly drawn. The labels for the ecosystems do not seem to correspond to where the ecosystems are although I understand that the authors found it difficult to place them on the actual systems. The map looks amateurish.

Figure 2 is a very difficult figure to understand at first glance. For example if mangrove leaves are a source (symbol is a cross) then why are they not deceasing in size similar to the other sources? In addition the symbols decreasing in size do not help comprehension of the figure. This information may be better shown in a table.

Figure 4 and 5. Could this be done better as a percentage column graph? From my comments on the discussion, I found it difficult to see the patterns the authors stated were there.

Figure 6, should either be deleted, put into a table or put into the supplementary material.

Discussion

In the first section (4.1) of the discussion the authors state (pg 11 Line 2-5) that the depleted 13C in seagrasses and macroalgae could be due to the different physical settings of the two creeks. The depleted 13C in seagrasses and macroalgae is also referred to in the results but I cannot see how this can be inferred by the associated figure. The explanation is plausible but I cannot see the evidence from the data the authors refer to. It would be easier to understand if this figure (fig.2) was represented in a table.

Section 4.2 states that mangrove derived material from Kinondo Creek greatly contributed to the sedimentary pool and moving seaward decreased its contribution until

dropping steeply in the coral reef. This is true for fatty acids (fig. 7) but I cannot see from the isotopes (fig. 4 and 5) how the authors came to this conclusion. In transect B (Kinondo Creek), mangrove derived material does contribute to the sediment although I would not state it contributed the most nor does it decrease from mangrove forests to seagrass beds, at some points it increased its contribution! However the authors are correct in that its contribution steeply drops off at the coral reef. The authors also state that the transect A, due to the influx of freshwater the export of mangrove derived material is further and indicates a significant contribution to the whole bay. Again this explanation is plausible and the authors place their explanation well within the known literature. But if Figure 4 and 5, which are suppose to represent this pattern, they do not show this. Considering that the authors state that there is a difference between the creeks, I cannot see a statistical significant difference between the two transects from the figures. However their explanation is plausible and I wonder if the wrong data is being shown? The fatty acids do confirm the authors explanation, not the isotopes.

Section 4.2 (pg 12, lines 7-9), the authors should include a line or two regarding seasonal changes in litter fall. Avicennia sp. in Brazil will lose their leaves only directly between seasons whilst Rhizophora sp. have leaf fall continuously over the seasons.

All other comments for the discussion are minor.

Minor comments

Pg 2, line 20. Hemminga et al. 1994 is a eminent paper but not a recent one.

Line 26. Wave power is not the correct word to use here.

Pg 3, line 6-10. Nitrogen isotopes are also used and should be mentioned here.

Line 23, A reference should be given for rainfall magnitude.

Pg 4, line 19. What is the depth of the coral reef?

Line 23. "unless low groundwater discharge" please re-write for clarity.

Line 24. What are the high flow rates?

Line 27. "ones" is not a scientific word.

Pg 5, line 2 "one" not a scientific word.

Line 8. "ones" not a scientific word.

Line 14. How much volume of sediment was collected in the cores?

Line 15. How were the seagrass leaves and macroalgae sampled, plucked?

Line 18. How were the samples kept cool and dark before arrival?

Line 27. What type of micro mill was it?

Pg 6, line 6. The equation could be presented much more clearly.

Line 10. I cannot understand what you did from this sentence.

Line 25. Why did you not also look at terrestrial sources, two lines in the introduction should provide some justification for not using terrestrial sources.

Pg 11, line 19. Do you mean the transects when you state "stations".

Pg 12, line 21. When was the timing of the other studies?

Line 22-25. I find this line confusing and had to read it several times to understand what the authors were meaning.

―――――――――――――――――――

---

## Referee Comment (RC2) · T. Meziane (Referee) · 28 Sep 2016

General comments:

This paper deals with the fate of the mangrove derived organic matter in relation with local geomorphological differences. Using elemental, isotopic and Fatty acids markers, the authors emphasize the combined role of the tide and the riverine water runoff in the distribution of the Mangrove organic matter. This paper is the latest of a long series of studies that characterised OM in Gazi Bay. The "plus" of this paper is the recording of FAs data and the fact that two seasons were sampled. Therefore, the main finding of this paper is the seasonal differences in term of export, which help to better understand the OM dynamic in the Bay; the combined control of tide and runoff is not something

exceptional as this is a general feature of estuarine mangrove and this must have been anticipated. The paper is well written and organised with however overstretching use of the fatty acid method. I have several concern, some I share with the other reviewer, I already read the comments, such as the statistical issues.

My main concern is on the manner how fatty acids were ascribed to sources is this study. The Fatty acid markers method, have evolved quiet a lot in the recent years. The use of FAs in a qualitative, at best semi-quantitative, manner (%) need some precautions when it comes to comparing them in living tissues and to extrapolate these relative contributions to "non living" matter in the environment. The conservative feature of these markers do not apply in sediment or POM and most of the fatty acids, at least most of those used in this study, cannot be ascribed solely to one particular source. It is probably not necessary to analyse microorganisms such as Bacteria but it would have been suitable to look at the composition of microalgae and terrestrial sources that can be brought by water draining. Indeed, relating 20:5w3 to red algae is not a good assumption when this FA is readily present in diatoms and/or other brown algae who seem to be dominant in this bay. The question is the big amount of brown algae (+ diatoms) with low % can be of less impact than red macroalgae with high % of 20:5w3 but low biomass? Also, 18:3w3 is also found in large amounts if terrestrial leaves and is more labile than LCFAs in sediments. 18:2w6 is very common in wastes and agriculture waters and we don't have indication about these possibilities in the Method's section. Also there in no indication about the seasonal changes that may affect the composition of sources which certainly can moderate here or there their relative contributions at the surface sediments as well as in the SPOM. A better knowledge on available sources and how their productions are impacted by seasonal patterns would have render this spatially restricted study to be less speculative in term on fatty acids evidences.

Other comments

Introduction P4L10 : typo : approaches P4L2 : it is important de say if it is a qualitative

or quantitative contribution

M&M : P 6 L18: Here and the entire paper, including tables and figures; the terminology of saturated Fatty acids is not properly defined : the is one "0" to much 23:0 instead of 23:00 and so on for all the paper.

P7 L4: I am puzzled by the transformation arcsine square root because % data are "transformed " (total 100%) which means that they have to be used as it is.

P7 L10 : using SIMPER to identify potential FAs is somehow wrong , SIMPER give you what are the FA that contribute the most to the similarity . A small contribution of a "specific" FA, say a Branched one for Bacteria, would be a enough to trace the OM and still, will not show up in the best five of the primer analysis. This practice adds confusion on the data that are % but discussed in a quantitative manner.

Results : P7 L15to L21: all comparisons need to be tested statistically P8 L5 the Bayesian model (SIAR) is may be not needed to see the contributions of the sources since there is no fractionation to correct.

P9 : L 25: Using 18:2w6+18:3w3 as tracers of seagrasses in zone full of mangrove is very risky.

Discussion :

P10 L26 : In this paragraph it will be useful to to discuss possible ocean inputs (seaward station) P13 L4-L5 : Speculative. P4L15 : 16:1w7/16:0 is certainly not an indicator of dino/diatoms and , 20:5w3/22ww3 is a diatom/dino marker (not the opposite) . Another reason to not ascribed 20:5w3 to red algae. P14 L25 : it is very speculative to relate a relative increase of bacterial FA (compare to what ? ) to an increase bacterial activity , at best it may show an increase in biomass but only if to compare the same site, for instance between season.

Figure 1 : the limit of the mangrove forest is not obvious in the map, please write Mkurumunji RIVER Fig 7 and 8 : besides all my comments on the use of markers, here

I would like to emphasize that Branched and 18:1w7 are surely tracers of bacteria, but one should complete the other and must not any more be added as it was done 20 years ago . there are many papers that show discrepancy between these two type of markers.

---

## Author Comment (AC1) · 17 Nov 2016

Comments The authors of this paper have investigated changes in mangrove out-welling to seagrass beds and coral reefs over seasons in Gazi Bay, Kenya. Mangrove forests are highly productive-ecosystems and have important implications for the exchange of organic material in the tropical coastal seascape. They have combined more traditional analyses (isotopes) with modern techniques (fatty acids) to explain the source contributions and spatial distribution of organic material across the bay. In addition they also investigated how physical factors such as tidal exchanges, river runoff and season may affect the exchange of organic material between different habitats. The authors state they have found significant differences in transport of organic mate-

rial from mangrove forest creeks to adjacent systems. This was primarily due to river runoff and tidal transport. However the river runoff creek was influenced strongly by the rainy season. Interestingly the authors found that macroalgae also had a contribution to organic material, this ecosystem is often ignored in the tropical seascape. The paper is of an interesting subject with new techniques applied. There is a dearth of data regarding the outwelling of mangroves with physical factors taken into account especially with respect to connectivity. It is quite well written and structured. The abstract and introduction are well done and flow nicely. There are minor comments on these sections. I admire the authors for doing such a through job on this study, they have tried to expand our understanding using new techniques in addition they must have worked very hard. It is not easy to complete fieldwork over two seasons and fatty acids are very labor intensive. However, I have some concerns regarding the methodology, results and major concerns (section 4.2) regarding the discussion.

Methodology No statistics were used to compare the isotopes sources. This makes it difficult to understand the patterns in the data the authors state. The changes the authors saw in the organic material sources did not seem statistically significant nor could was it shown in the figures. A statistical test would resolve this.

R: We agree with the reviewer that statistical tests were needed to support our results and discussion. Consequently, we have performed permutational multivariate ANOVA (PERMANOVA) both on d13C and C/N values of organic matter sources and on the outputs of the Bayesian models (lower and upper limit of the credibility intervals, mode and mean). In both cases, the outcomes of the analysis highlighted that the patterns described in the text were significant, hence the main message of the manuscript was unchanged. PERMANOVA on the Bayesian mixing models will be showed in a new (now Table 1).

Results The figures depicting the results are numerous and difficult to understand. Several of the tables/figures could go into the supplementary section (even though there is already data in this section), other figures need to be edited for clarity and

others could be drastically improved by a different method of illustrating them. Several issues in the discussion may be due to the difficulty in understanding the figures. In addition I and I think others would find it easier if at some points the authors specified which creek applied to transects A and B, it was annoying to keep on referring to the methodology to understand which was which.

R: We thank the reviewer for his/her constructive comments. Consequently, we will made several changes in tables and figures (described in more detail below). Moreover, throughout the manuscript, we will refer to the creek name, together with the transect letter.

For example, section 3.1.2, line 7- I am not sure if "often" is the correct word to use. They seemed to be almost always overlapping.

R: We will change "often" with "overall".

Table 1 and 2 could go in the supplementary material. Table 1 especially should be in the supplementary material.

R: Both Table 1 and 2 will be moved in supplementary materials (now Suppl. 1 and 2).

Figure 1 is badly drawn. The labels for the ecosystems do not seem to correspond to where the ecosystems are although I understand that the authors found it difficult to place them on the actual systems. The map looks amateurish.

R: We have changed Figure 1 to improve clarity. The names of the stations have been placed in the right position. "Creek" and "river" have been added to the watercourse names. Moreover, we have drawn both coastline and transects.

Figure 2 is a very difficult figure to understand at first glance. For example if mangrove leaves are a source (symbol is a cross) then why are they not deceasing in size similar to the other sources? In addition the symbols decreasing in size do not help comprehension of the figure. This information may be better shown in a table.

R: We agree with the reviewer 1 that Figure 2 is not clear as it was drawn in the previous draft. Nevertheless, we think it is more appropriated to keep a figure for representing isotopic and elemental data, rather than drawing a table, because the scatter plot draws together both variables and allows to see reciprocal trends at a first glance. Hence, we have changed the figure to make it clearer and more understandable. The sources from the different stations have been drawn in different colors instead of different size. There is only one symbol for mangrove leaves (the black cross) because they were collected only in the mangrove stations, as we will specify in the manuscript.

Figure 4 and 5. Could this be done better as a percentage column graph? From my comments on the discussion, I found it difficult to see the patterns the authors stated were there.

R: Figures 4 and 5 represent the probability distributions that a certain source contribute to the two abiotic compartments studied: SOM and SPOM. Hence, it is important to show the entire credibility interval, indicating both the modes and the upper and lower limits of the 95% credibility interval. Anyway, we agree with the reviewer that the patterns described in the text are not always evident from the figures. To address this issue, we will perform a statistical analysis (PERMANOVA) and change the whole paragraph to make clearer the description of the patterns.

Figure 6, should either be deleted, put into a table or put into the supplementary material.

R: Figure 6 has been deleted.

Discussion In the first section (4.1) of the discussion the authors state (pg 11 Line 2-5) that the depleted 13C in seagrasses and macroalgae could be due to the different physical settings of the two creeks. The depleted 13C in seagrasses and macroalgae is also referred to in the results but I cannot see how this can be inferred by the associated figure. The explanation is plausible but I cannot see the evidence from the data the authors refer to. It would be easier to understand if this figure (fig.2) was represented

in a table.

R: We agree with the reviewer that figure 2 is not clear enough. Now we we have modified the figure to make it clearer, and the trends described in the text will be also changed to be more intelligible. In the right panels (transect B corresponding to Kinondo creek), $\delta$13C values of macroalgae and seagrasses from the mangrove stations (black diamonds and circle respectively) are more negative (macroalgae $\sim$ -25‰ $seagrasses\sim$ -20‰ than those represented in the left panels (macroalgae $\sim$ -20‰ $seagrasses\sim$ -15‰. Moreover, PERMANOVA confirmed that these differences were significant.

Section 4.2 states that mangrove derived material from Kinondo Creek greatly contributed to the sedimentary pool and moving seaward decreased its contribution until dropping steeply in the coral reef. This is true for fatty acids (fig. 7) but I cannot see from the isotopes (fig. 4 and 5) how the authors came to this conclusion. In transect B (Kinondo Creek), mangrove derived material does contribute to the sediment although I would not state it contributed the most nor does it decrease from mangrove forests to seagrass beds, at some points it increased its contribution! However the authors are correct in that its contribution steeply drops off at the coral reef. The authors also state that the transect A, due to the influx of freshwater the export of mangrove derived material is further and indicates a significant contribution to the whole bay. Again this explanation is plausible and the authors place their explanation well within the known literature. But if Figure 4 and 5, which are suppose to represent this pattern, they do not show this. Considering that the authors state that there is a difference between the creeks, I cannot see a statistical significant difference between the two transects from the figures. However their explanation is plausible and I wonder if the wrong data is being shown? The fatty acids do confirm the authors explanation, not the isotopes.

R: Thanks to the reviewer's comment, we have revised the whole isotopic dataset and actually there were a few typos: mean $\delta$13C ($\pm$ ds) of macroalgae from the intertidal area along transect A (-20.08 $\pm$ 0.66 ‰ instead of -20.63 $\pm$ 1.41 ‰ and along transect

B (-21.79 ± 1.44 ‰ instead of -20.12 ± 0.89 ‰ and the mean $\delta$13C (± ds) of sea-grasses from the mangrove area along transect B (-20.23 ± 1.74 ‰ instead of -19.34 ± 2.83 ‰. Nevertheless, the patterns of the contribution of organic matter sources to SOM and SPOM did not change significantly. We agree with the reviewer that the patterns described in the text are not striking from figures 4 and 5. Hence, we run PER-MANOVA to test for differences among seasons, transects and stations and results highlighted significant differences in all cases. Pair-wise tests among stations were always significant too. Moreover, we will modify this paragraph, softening the discussion about mixing models outcomes and focusing more on fatty acid results. Section 4.2 (pg 12, lines 7-9), the authors should include a line or two regarding seasonal changes in litter fall. Avicennia sp. in Brazil will lose their leaves only directly between seasons whilst Rhizophora sp. have leaf fall continuously over the seasons.

R: We are grateful to the reviewer for this interesting comment. We will add a few lines on the seasonal variability in litterfall in Gazi Bay.

All other comments for the discussion are minor. Minor comments Pg 2, line 20. Hemminga et al. 1994 is a eminent paper but not a recent one.

R: We will remove Hemminga et al. 1994.

Line 26. Wave power is not the correct word to use here.

R: We will change "wave power" with "wave action".

Pg 3, line 6-10. Nitrogen isotopes are also used and should be mentioned here.

R: d15N is a powerful tool to identify diet and trophic levels of consumers in natural ecosystems, rather than to trace organic matter sources. This is because tracing organic nitrogen food sources is complicated by trophic enrichment, and then it is rarely used to this end. Consequently, we think that mentioning nitrogen stable isotopes here would be misleading.

Line 23, A reference should be given for rainfall magnitude.

R: We will add a reference.

Pg 4, line 19. What is the depth of the coral reef?

R: We will add this information.

Line 23. "unless low groundwater discharge" please re-write for clarity.

R: We will rephrase the sentence to improve clarity.

Line 24. What are the high flow rates?

R: We will add this information.

Line 27. "ones" is not a scientific word.

R: We will change "ones" with "flows".

Pg 5, line 2 "one" not a scientific word.

R: We will rephrase the sentence to remove the word "one".

Line 8. "ones" not a scientific word.

R: We will rephrase the sentence to remove the word "ones".

Line 14. How much volume of sediment was collected in the cores?

R: The cores were approximately half-filled but only the first 5 cm were used for the analysis, as specified in the text.

Line 15. How were the seagrass leaves and macroalgae sampled, plucked?

R: We agree that the sentence was not clear. We did not collect the leaves, but the whole shoots (seagrass) and thalli (macroalgae). Hence, we will change the sentence to specify this.

Line 18. How were the samples kept cool and dark before arrival?

R: We will add "in a cool box".

Line 27. What type of micro mill was it?

R: We will add the model of the micro mill.

Pg 6, line 6. The equation could be presented much more clearly.

R: The equation will be centered in the page and spaces will be inserted to improve clarity and readability.

Line 10. I cannot understand what you did from this sentence.

R: A more detailed description of the lipid extraction method will be provided.

Line 25. Why did you not also look at terrestrial sources, two lines in the introduction should provide some justification for not using terrestrial sources.

R: Mangroves are the dominant terrestrial sources in the area as highlighted by previous research, but agricultural runoff from the sugar plantations surrounding the forest might also flow into the watercourses. We will specify both information in the Introduction section and also we will take into account the potential contribution of agricultural runoff through the fatty acid approach, as suggested by reviewer 2.

Pg 11, line 19. Do you mean the transects when you state "stations".

R: We agree that the sentence was not clear because we meant stations, not transects. Now we will change the sentence to clarify our point of discussion.

Pg 12, line 21. When was the timing of the other studies?

R: The timing of sampling is not specified in Bouillon et al (2007). In contrast, Hemminga et al (1994) sampled at both ebb and flood tide. The spatial gradient of the SPOM d13C found by Hemminga et al (1994) at ebb tide was similar to that found in this study. Hence, we will mention only Hemminga et al 1994 highlighting the comparable trend.

Line 22-25. I find this line confusing and had to read it several times to understand what the authors were meaning.

R: In this sentence, we meant that, despite the buffering role of seagrass beds in preventing a direct connection between mangrove and oceanic waters, we have inferred that high mangrove export coupled with high rate of water exchange at spring ebb tide has favoured the outwelling of suspended mangrove material up to the coral reef inner area. We will simplify the sentence to improve understandability.

Please also note the supplement to this comment:
http://www.biogeosciences-discuss.net/bg-2016-302/bg-2016-302-AC1-supplement.pdf

**Fig. 1.** figure 1

[Figure]

[Figure]

Fig. 2. figure 2

[Figure]

**Fig. 3.** figure 3

**Fig. 4.** figure 4

[Figure]

**Fig. 5.** figure 5

---

## Author Comment (AC2) · 17 Nov 2016

General comments: This paper deals with the fate of the mangrove derived organic matter in relation with local geomorphological differences. Using elemental, isotopic and Fatty acids markers, the authors emphasize the combined role of the tide and the riverine water runoff in the distribution of the Mangrove organic matter. This paper is the latest of a long series of studies that characterised OM in Gazi Bay. The "plus" of this paper is the recording of FAs data and the fact that two seasons were sampled. Therefore, the main finding of this paper is the seasonal differences in term of export, which help to better understand the OM dynamic in the Bay; the combined control of tide and runoff is not something exceptional as this is a general feature of estuarine

mangrove and this must have been anticipated.

R: We agree that the combined effect of runoff and tide, together with rainfall and wave action is a general feature of tropical estuarine systems. Indeed, this point was already mentioned in the previous version of the manuscript. Now we will strengthened further this point.

The paper is well written and organised with however overstretching use of the fatty acid method. I have several concern, some I share with the other reviewer, I already read the comments, such as the statistical issues.

R: Following the reviewer's comments, we will dampen the use of the fatty acid approach (see specific comments below). As for the statistical analysis, in the revised manuscript we will carry out permutational analysis of variance (PERMANOVA) to test for spatio-temporal differences in isotopic and elemental signatures of organic matter sources and in their contribution to SOM and SPOM, using the outcomes of the Bayesian mixing models (lower and upper limit of the credibility intervals, mode and mean) as variables.

My main concern is on the manner how fatty acids were ascribed to sources is this study. The Fatty acid markers method, have evolved quiet a lot in the recent years. The use of FAs in a qualitative, at best semi-quantitative, manner (%) need some precautions when it comes to comparing them in living tissues and to extrapolate these relative contributions to "non living" matter in the environment. The conservative feature of these markers do not apply in sediment or POM and most of the fatty acids, at least most of those used in this study, cannot be ascribed solely to one particular source. It is probably not necessary to analyse microorganisms such as Bacteria but it would have been suitable to look at the composition of microalgae and terrestrial sources that can be brought by water draining. Indeed, relating 20:5w3 to red algae is not a good assumption when this FA is readily present in diatoms and/or other brown algae who seem to be dominant in this bay. The question is the big amount of brown algae (+

diatoms) with low % can be of less impact than red macroalgae with high % of 20:5w3 but low biomass? Also, 18:3w3 is also found in large amounts if terrestrial leaves and is more labile than LCFAs in sediments. 18:2w6 is very common in wastes and agriculture waters and we don't have indication about these possibilities in the Method's section. Also there in no indication about the seasonal changes that may affect the composition of sources which certainly can moderate here or there their relative contributions at the surface sediments as well as in the SPOM. A better knowledge on available sources and how their productions are impacted by seasonal patterns would have render this spatially restricted study to be less speculative in term on fatty acids evidences.

R: We are very grateful to the reviewer for the stimulating comments. Now we will soften the use of many FA tracers, using much more caution than in the previous version of the manuscript. In particular, now we will treat 18:3 n3 as a combined marker of mangroves and seagrasses and 18:2 n6 as a combined tracer of seagrasses and agricultural runoff from the sugar plantations diffused around the bay (we will add this information also in the Method's section). In addition, for both fatty acids, we will take into account the potential lability due to decomposition as a discussion point for explaining their low relative abundance in SOM and SPOM. As for 20:5 n3, SIMPER results highlighted high 20:5 n3 content in red algae, and not in brown algae, hence, we treated this FA as a combined tracer of diatoms and red algae. Moreover, as suggested by the reviewer 1, we will discuss the seasonal variability of mangrove litterfall to explain the seasonal patterns recorded in this study. Seasonal and spatial variability was detected also in bacterial biomarker and will be discussed in the manuscript.

Other comments Introduction P4L10 : typo : approaches

R: The typo will be corrected.

P4L2 : it is important de say if it is a qualitative or quantitative contribution

R: We will rephrase the sentence pointing out that the contribution of dominant pri-

mary producers to sedimentary and suspended particulate organic matter pools was assessed based on quantitative (isotope mixing models) and semi-quantitative (fatty acids) approaches.

M&M : P 6 L18: Here and the entire paper, including tables and figures; the terminology of saturated Fatty acids is not properly defined : the is one "0" to much 23:0 instead of 23:00 and so on for all the paper.

R: We will correct the FA nomenclature through the paper.

P7 L4: I am puzzled by the transformation arcsine square root because % data are "transformed " (total 100%) which means that they have to be used as it is.

R: Proportional fatty acid (percentage of total FAME) data require transformation to meet the assumption of multivariate normality (Budge et al. 2006). The arcsine square root transformation is commonly used for proportional fatty acid data (e.g. Iverson, 2009; Thiemann et al., 2011; Raymond et al., 2014).

Budge, Suzanne M., Sara J. Iverson, and Heather N. Koopman. "Studying trophic ecology in marine ecosystems using fatty acids: a primer on analysis and interpretation." Marine Mammal Science 22.4 (2006): 759-801.

Iverson, Sara J. "Tracing aquatic food webs using fatty acids: from qualitative indicators to quantitative determination." Lipids in Aquatic Ecosystems. Springer New York, 2009. 281-308.

Thiemann, Gregory W., et al. "Individual patterns of prey selection and dietary specialization in an Arctic marine carnivore." Oikos 120.10 (2011): 1469-1478.

Raymond, Wendel W., Alexander T. Lowe, and Aaron WE Galloway. "Degradation state of algal diets affects fatty acid composition but not size of red urchin gonads." Marine Ecology Progress Series 509 (2014): 213-225.

P7 L10 : using SIMPER to identify potential FAs is somehow wrong , SIMPER give

you what are the FA that contribute the most to the similarity . A small contribution of a "specific" FA, say a Branched one for Bacteria, would be a enough to trace the OM and still, will not show up in the best five of the primer analysis. This practice adds confusion on the data that are % but discussed in a quantitative manner.

R: SIMPER is a common routine to identify fatty acids that contribute to similarity within groups and dissimilarity between groups. We agree with the reviewer that this approach can be misleading in some cases. However, in this study, we used SIMPER only to identify the main fatty acids that characterized the primary producers sampled in the area (accordingly to Kelly and Scheibling, 2012). Then, the identified FAs were used as indicators of specific primary producer-derived organic matter in the abiotic compartments, assuming that the relative abundance of specific FA indicators in sedimentary or suspended organic matter will be proportional to the contribution of the correspondent primary producer. The contribution of other potential organic matter sources to SOM and SPOM, as Bacteria, was assessed using the biomarkers published in literature. To clarify better the aims and the results of this statistical approach, we will specify better the objectives of the SIMPER analysis highlighting that this approach is used to identify the FAs that contributed more to the similarity within and dissimilarity between primary producer groups (Clarke and Warwick, 2001). Therefore, we will specify that these FAs are used only as indicators of specific primary producer-derived organic matter for sedimentary and suspended particulate material characterisation, together with those reported in current literature. To add clarity to these information, we will include also the dissimilarities between groups in the table (Suppl. 1).

Kelly, Jennifer R., and Robert E. Scheibling. "Fatty acids as dietary tracers in benthic food webs." Marine Ecology Progress Series 446 (2012): 1-22.

Clarke K.R., Warwick R.M. "Change in marine communities: an approach to statistical analysis and interpretation." Plymouth, UK: Primer-E (2001).

Results : P7 L15to L21: all comparisons need to be tested statistically

R: All comparisons will be tested through permutational analysis of variance (PER-MANOVA).

P8 L5 the Bayesian model (SIAR) is may be not needed to see the contributions of the sources since there is no fractionation to correct.

R: Bayesian models are valid tools to assess the contribution of sources to consumers (in this case SOM and SPOM) even regardless of fractionation value. In this study, we analysed the contribution of organic matter sources to sediment and particulate compartment assuming that their isotopic compositions remained unchanged after their incorporation in SOM and SPOM accordingly to Gonneea et al. (2004). Now we will specify this the manuscript.

P9 : L 25: Using 18:2w6+18:3w3 as tracers of seagrasses in zone full of mangrove is very risky.

R: We agree and now we will be much more cautious in using FAs as tracers of primary producers. The whole paragraph will be rephrased, considering 18:3 n3 as a tracer of both seagrasses and mangroves, 18:2 n6 as a tracer of both seagrasses and agricultural runoff and 20:5 n3 as a tracer of both diatoms and red macroalgae.

Discussion : P10 L26 : In this paragraph it will be useful to discuss possible ocean inputs (seaward station)

R: The influence of oceanic input in the seaward station, in terms of influence of oceanic dissolved inorganic carbon on the carbon isotopic signature of primary producers, was discussed in the subsequent sentence. In particular, we stated that "A similar enrichment was already observed in Gazi Bay and other tropical areas (Hemminga et al., 1994; Lugendo et al., 2007) and mirrors changes in d13CDIC (Alongi, 2014; Maher et al., 2013). d13CDIC is typically more negative close to mangroves as a result of the intense localized mineralization of mangrove detritus (Bouillon et al., 2007) and increases seaward due to the increased contribution of oceanic DIC, whose d13C is

typically around 0‰ (Bouillon et al., 2008)". If necessary, further details will be added.

P13 L4-L5 : Speculative.

R: We will remove this sentence.

P4L15 : 16:1w7/16:0 is certainly not an indicator of dino/diatoms and , 20:5w3/22ww3 is a diatom/dino marker (not the opposite) . Another reason to not ascribed 20:5w3 to red algae.

R: We thank the reviewer for pointing out this mistake. Now we will correct it and change the sentence giving more importance to 20:5 n3/22:6 n3 as a diatom/dinoflagellate marker. We have decided to eliminate the mention to 16:1w7/16:0 because there was not a univocal pattern.

P14 L25 : it is very speculative to relate a relative increase of bacterial FA (compare to what ? ) to an increase bacterial activity , at best it may show an increase in biomass but only if to compare the same site, for instance between season.

R: We are grateful to the reviewer for this comment. We will change "greater benthic mineralization" with "high bacterial biomass".

Figure 1 : the limit of the mangrove forest is not obvious in the map, please write Mkurumunji RIVER

R: We agree that Figure 1 was not clear, as it was pointing out also by reviewer 1. Then we have changed it to improve clarity. The names of the stations have been placed in the right position. "Creek" and "river" have been added to the watercourse names. Moreover, we have drawn the transects and the coastline to improve the identification of the limits of the mangrove forest.

Fig 7 and 8 : besides all my comments on the use of markers, here I would like to emphasize that Branched and 18:1w7 are surely tracers of bacteria, but one should complete the other and must not any more be added as it was done 20 years ago .

there are many papers that show discrepancy between these two type of markers.

R: We have modified figures 7 and 8 (now 6 and 7) showing individually the two bacterial markers (branched and 18:1 n7) . Moreover, following previous comments of the reviewer, also 18:3 n3 and 18:2 n6 have been indicated individually.

Please also note the supplement to this comment:
http://www.biogeosciences-discuss.net/bg-2016-302/bg-2016-302-AC2-supplement.pdf

[Figure]

**Fig. 1.** figure 6

[Figure]

**Fig. 2.** figure 7

---

## Author Response (AR1)

**Comments**

The authors of this paper have investigated changes in mangrove outwelling to seagrass beds and coral reefs over seasons in Gazi Bay, Kenya. Mangrove forests are highly productive-ecosystems and have important implications for the exchange of organic material in the tropical coastal seascape. They have combined more traditional analyses (isotopes) with modern techniques (fatty acids) to explain the source contributions and spatial distribution of organic material across the bay. In addition they also investigated how physical factors such as tidal exchanges, river runoff and season may affect the exchange of organic material between different habitats.

The authors state they have found significant differences in transport of organic material from mangrove forest creeks to adjacent systems. This was primarily due to river runoff and tidal transport. However the river runoff creek was influenced strongly by the rainy season. Interestingly the authors found that macroalgae also had a contribution to organic material, this ecosystem is often ignored in the tropical seascape.

The paper is of an interesting subject with new techniques applied. There is a dearth of data regarding the outwelling of mangroves with physical factors taken into account especially with respect to connectivity. It is quite well written and structured. The abstract and introduction are well done and flow nicely. There are minor comments on these sections. I admire the authors for doing such a through job on this study, they have tried to expand our understanding using new techniques in addition they must have worked very hard. It is not easy to complete fieldwork over two seasons and fatty acids are very labor intensive. However, I have some concerns regarding the methodology, results and major concerns (section 4.2) regarding the discussion.

**Methodology**

No statistics were used to compare the isotopes sources. This makes it difficult to understand the patterns in the data the authors state. The changes the authors saw in the organic material sources did not seem statistically significant nor could was it shown in the figures. A statistical test would resolve this.

R: We agree with the reviewer that statistical tests were needed to support our results and discussion. Consequently, we have performed permutational multivariate ANOVA (PERMANOVA) both on $d^{13}C$ and C/N values of organic matter sources and on the outputs of the Bayesian models (lower and upper limit of the credibility intervals, mode and mean). In both cases, the outcomes of the analysis highlighted that the patterns described in the text were significant, hence the main message of the manuscript was unchanged. PERMANOVA on the Bayesian mixing models will be showed in a new (now Table 1).

**Results**

The figures depicting the results are numerous and difficult to understand. Several of the tables/figures could go into the supplementary section (even though there is already data in this section), other figures need to be edited for clarity and others could be drastically improved by a different method of illustrating them. Several issues in the discussion may be due to the difficulty in understanding the figures. In addition I and I think others would find it easier if at some points the authors specified which creek applied to transects A and B, it was annoying to keep on referring to the methodology to understand which was which.

R: We thank the reviewer for his/her constructive comments. Consequently, we have made several changes in tables and figures (described in more detail below). Moreover, throughout the manuscript, we refer to the creek name, together with the transect letter.

For example, section 3.1.2, line 7- I am not sure if "often" is the correct word to use. They seemed to be almost always overlapping.

R: We have changed "often" with "overall". P. 9 L. 2

Table 1 and 2 could go in the supplementary material. Table 1 especially should be in the supplementary material.

R: Both Table 1 and 2 are in supplementary materials (Suppl. 1 and 2).

Figure 1 is badly drawn. The labels for the ecosystems do not seem to correspond to where the ecosystems are although I understand that the authors found it difficult to place them on the actual systems. The map looks amateurish.

R: We have changed Figure 1 to improve clarity. The names of the stations have been placed in the right position. "Creek" and "river" have been added to the watercourse names. Moreover, we have drawn both coastline and transects.

Figure 2 is a very difficult figure to understand at first glance. For example if mangrove leaves are a source (symbol is a cross) then why are they not deceasing in size similar to the other sources? In addition the symbols decreasing in size do not help comprehension of the figure. This information may be better shown in a table.

R: We agree with the reviewer 1 that Figure 2 is not clear as it was drawn in the previous draft. Nevertheless, we think it is more appropriated to keep a figure for representing isotopic and elemental data, rather than drawing a table, because the scatter plot draws together both variables and allows to see reciprocal trends at a first glance. Hence, we have changed the figure to make it clearer and more understandable. The sources from the different stations have been drawn in different colors instead of different size. There is only one symbol for mangrove leaves (the black cross) because they were collected only in the mangrove stations, as we will specify in the manuscript.

Figure 4 and 5. Could this be done better as a percentage column graph? From my comments on the discussion, I found it difficult to see the patterns the authors stated were there.

R: Figures 4 and 5 represent the probability distributions that a certain source contribute to the two abiotic compartments studied: SOM and SPOM. Hence, it is important to show the entire credibility interval, indicating both the modes and the upper and lower limits of the 95% credibility interval. Anyway, we agree with the reviewer that the patterns described in the text are not always evident from the figures. To address this issue, we have performed a statistical analysis (PERMANOVA) and changed the whole paragraph to make clearer the description of the patterns.

Figure 6, should either be deleted, put into a table or put into the supplementary material.

R: Figure 6 has been deleted.

**Discussion**
In the first section (4.1) of the discussion the authors state (pg 11 Line 2-5) that the depleted 13C in seagrasses and macroalgae could be due to the different physical settings of the two creeks. The depleted 13C in seagrasses and macroalgae is also referred to in the results but I cannot see how this can be inferred

by the associated figure. The explanation is plausible but I cannot see the evidence from the data the authors refer to. It would be easier to understand if this figure (fig.2) was represented in a table.

R: We agree with the reviewer that figure 2 is not clear enough. Now we have modified the figure to make it clearer, and the trends described in the text have been also changed to be more intelligible. In the right panels (transect B corresponding to Kinondo creek), $\delta^{13}$C values of macroalgae and seagrasses from the mangrove stations (black diamonds and circle respectively) are more negative (macroalgae ~ -25‰; seagrasses ~ -20‰) than those represented in the left panels (macroalgae ~ -20‰; seagrasses ~ -15‰). Moreover, PERMANOVA confirmed that these differences were significant.

Section 4.2 states that mangrove derived material from Kinondo Creek greatly contributed to the sedimentary pool and moving seaward decreased its contribution until dropping steeply in the coral reef. This is true for fatty acids (fig. 7) but I cannot see from the isotopes (fig. 4 and 5) how the authors came to this conclusion. In transect B (Kinondo Creek), mangrove derived material does contribute to the sediment although I would not state it contributed the most nor does it decrease from mangrove forests to seagrass beds, at some points it increased its contribution! However the authors are correct in that its contribution steeply drops off at the coral reef. The authors also state that the transect A, due to the influx of freshwater the export of mangrove derived material is further and indicates a significant contribution to the whole bay. Again this explanation is plausible and the authors place their explanation well within the known literature. But if Figure 4 and 5, which are suppose to represent this pattern, they do not show this. Considering that the authors state that there is a difference between the creeks, I cannot see a statistical significant difference between the two transects from the figures. However their explanation is plausible and I wonder if the wrong data is being shown? The fatty acids do confirm the authors explanation, not the isotopes.

R: Thanks to the reviewer's comment, we have revised the whole isotopic dataset and actually there were a few typos: mean $\delta^{13}$C (± ds) of macroalgae from the intertidal area along transect A (-20.08 ± 0.66 ‰ instead of -20.63 ± 1.41 ‰) and along transect B (-21.79 ± 1.44 ‰ instead of -20.12 ± 0.89 ‰) and the mean $\delta^{13}$C (± ds) of seagrasses from the mangrove area along transect B (-20.23 ± 1.74 ‰ instead of -19.34 ± 2.83 ‰). Nevertheless, the patterns of the contribution of organic matter sources to SOM and SPOM did not change significantly. We agree with the reviewer that the patterns described in the text are not striking from figures 4 and 5. Hence, we have run PERMANOVA to test for differences among seasons, transects and stations and results highlighted significant differences in all cases. Pair-wise tests among stations were always significant too. Moreover, we have modified this paragraph, softening the discussion about mixing models outcomes and focusing more on fatty acid results.

Section 4.2 (pg 12, lines 7-9), the authors should include a line or two regarding seasonal changes in litter fall. Avicennia sp. in Brazil will lose their leaves only directly between seasons whilst Rhizophora sp. have leaf fall continuously over the seasons.

R: We are grateful to the reviewer for this interesting comment. We have added a few lines on the seasonal variability in litterfall in Gazi Bay. P. 14 L. 5-7

All other comments for the discussion are minor.
**Minor comments**
Pg 2, line 20. Hemminga et al. 1994 is a eminent paper but not a recent one.

R: We have removed Hemminga et al. 1994.

Line 26. Wave power is not the correct word to use here.

R: We have changed "wave power" with "wave action". P. 2 L. 27

Pg 3, line 6-10. Nitrogen isotopes are also used and should be mentioned here.

R: d$^{15}$N is a powerful tool to identify diet and trophic levels of consumers in natural ecosystems, rather than to trace organic matter sources. This is because tracing organic nitrogen food sources is complicated by trophic enrichment, and then it is rarely used to this end. Consequently, we think that mentioning nitrogen stable isotopes here would be misleading.

Line 23, A reference should be given for rainfall magnitude.

R: We have added a reference. P. 3 L. 25

Pg 4, line 19. What is the depth of the coral reef?

R: We have added this information. P. 4 L. 24

Line 23. "unless low groundwater discharge" please re-write for clarity.

R: We have rephrased the sentence to improve clarity. P. 4 L. 27

Line 24. What are the high flow rates?

R: We have added this information. P. 5 L. 2

Line 27. "ones" is not a scientific word.

R: We have changed "ones" with "flows". P. 5 L. 5

Pg 5, line 2 "one" not a scientific word.

R: We have rephrased the sentence to remove the word "one". P. 5 L. 5-8

Line 8. "ones" not a scientific word.

R: We have rephrased the sentence to remove the word "ones". P. 5 L. 12-15

Line 14. How much volume of sediment was collected in the cores?

R: The cores were approximately half-filled but only the first 5 cm were used for the analysis, as specified in the text. P. 6 L. 2-3

Line 15. How were the seagrass leaves and macroalgae sampled, plucked?

R: We agree that the sentence was not clear. We did not collect the leaves, but the whole shoots (seagrass) and thalli (macroalgae). Hence, we have changed the sentence to specify this. P. 5 L. 21-23

Line 18. How were the samples kept cool and dark before arrival?

R: We have added "in a cool box". P. 5 L. 25

Line 27. What type of micro mill was it?

R: We have added the model of the micro mill. P. 6 L. 8; 11

Pg 6, line 6. The equation could be presented much more clearly.

R: The equation has been centered in the page and spaces will be inserted to improve clarity and readability. P. 6 L. 15

Line 10. I cannot understand what you did from this sentence.

R: A more detailed description of the lipid extraction method has been provided. P. 6 L. 17-23

Line 25. Why did you not also look at terrestrial sources, two lines in the introduction should provide some justification for not using terrestrial sources.

R: Mangroves are the dominant terrestrial sources in the area as highlighted by previous research, but agricultural runoff from the sugar plantations surrounding the forest might also flow into the watercourses. We have specified both information: the former in the Introduction section (P. 4 L.11) and the latter in the Materials and methods section (P. 5 L.2-3). Moreover, we have also taken into account the potential contribution of agricultural runoff through the fatty acid approach, as suggested by reviewer 2.

Pg 11, line 19. Do you mean the transects when you state "stations".

R: We agree that the sentence was not clear because we meant stations, not transects. Now we have changed the sentence to clarify our point of discussion.  P. 13 L. 8-12

Pg 12, line 21. When was the timing of the other studies?

R: The timing of sampling is not specified in Bouillon et al (2007). In contrast, Hemminga et al (1994) sampled at both ebb and flood tide. The spatial gradient of the SPOM $d^{13}C$ found by Hemminga et al (1994) at ebb tide was similar to that found in this study. Hence, we mentioned only Hemminga et al 1994 highlighting the comparable trend. P. 14 L. 19-20

Line 22-25. I find this line confusing and had to read it several times to understand what the authors were meaning.

R: We agree with the reviewer that the sentence was confusing. Now we have changed the sentence as follows: "Despite the buffering role of seagrass beds in preventing a direct connection between mangrove and oceanic waters, we infer that high mangrove export coupled with high rate of water exchange at spring ebb tide has favoured the outwelling of suspended mangrove material up to the coral reef inner area". P. 14 L. 22-25

This paper deals with the fate of the mangrove derived organic matter in relation with local geomorphological differences. Using elemental, isotopic and Fatty acids markers, the authors emphasize the combined role of the tide and the riverine water runoff in the distribution of the Mangrove organic matter. This paper is the latest of a long series of studies that characterised OM in Gazi Bay. The "plus" of this paper is the recording of FAs data and the fact that two seasons were sampled. Therefore, the main finding of this paper is the seasonal differences in term of export, which help to better understand the OM dynamic in the Bay; the combined control of tide and runoff is not something exceptional as this is a general feature of estuarine mangrove and this must have been anticipated.

R: We agree that the combined effect of runoff and tide, together with rainfall and wave action is a general feature of tropical estuarine systems. Indeed, this point was already mentioned in the previous version of the manuscript. Now we have strengthened further this point. P. 2 L. 26 - P. 3 L. 6

The paper is well written and organised with however overstretching use of the fatty acid method.
I have several concern, some I share with the other reviewer, I already read the comments, such as the statistical issues.

R: Following the reviewer's comments, we have dampened the use of the fatty acid approach (see specific comments below). As for the statistical analysis we have carried out permutational analysis of variance (PERMANOVA) to test for spatio-temporal differences in isotopic and elemental signatures of organic matter sources and in their contribution to SOM and SPOM, using the outcomes of the Bayesian mixing models (lower and upper limit of the credibility intervals, mode and mean) as variables.

My main concern is on the manner how fatty acids were ascribed to sources is this study. The Fatty acid markers method, have evolved quiet a lot in the recent years. The use of FAs in a qualitative, at best semi-quantitative, manner (%) need some precautions when it comes to comparing them in living tissues and to extrapolate these relative contributions to "non living" matter in the environment. The conservative feature of these markers do not apply in sediment or POM and most of the fatty acids, at least most of those used in this study, cannot be ascribed solely to one particular source. It is probably not necessary to analyse microorganisms such as Bacteria but it would have been suitable to look at the composition of microalgae and terrestrial sources that can be brought by water draining. Indeed, relating 20:5w3 to red algae is not a good assumption when this FA is readily present in diatoms and/or other brown algae who seem to be dominant in this bay. The question is the big amount of brown algae (+ diatoms) with low % can be of less impact than red macroalgae with high % of 20:5w3 but low biomass? Also, 18:3w3 is also found in large amounts if terrestrial leaves and is more labile than LCFAs in sediments. 18:2w6 is very common in wastes and agriculture waters and we don't have indication about these possibilities in the Method's section. Also there in no indication about the seasonal changes that may affect the composition of sources which certainly can moderate here or there their relative contributions at the surface sediments as well as in the SPOM. A better knowledge on available sources and how their productions are impacted by seasonal patterns would have render this spatially restricted study to be less speculative in term on fatty acids evidences.

R: We are very grateful to the reviewer for the stimulating comments. Now we have softened the use of many FA tracers, using much more caution than in the previous version of the manuscript. In particular, now we have treated 18:3 n3 as a combined marker of mangroves and seagrasses and 18:2 n6 as a combined tracer of seagrasses and agricultural runoff from the sugar plantations diffused around the bay (we have also added this information also in the Method's section). In addition, for both fatty acids, we have taken into account the potential lability due to decomposition as a discussion point for explaining their low relative abundance in SOM and SPOM. As for 20:5 n3, SIMPER results highlighted high 20:5 n3 content in red algae, and not in brown algae, hence, we have treated this FA as a combined tracer of diatoms and red algae. Moreover, as suggested by the reviewer 1, we have discussed the seasonal variability of mangrove litterfall to explain the seasonal patterns recorded in this study. Seasonal and spatial variability was detected also in bacterial biomarker and has been discussed in the manuscript.

Other comments
Introduction P4L10 : typo : approaches

R: The typo has been corrected.

P4L2 : it is important de say if it is a qualitative or quantitative contribution

R: We have rephrased the sentence pointing out that the contribution of dominant primary producers to sedimentary and suspended particulate organic matter pools was assessed based on quantitative (isotope mixing models) and semi-quantitative (fatty acid profiles) approaches. P. 4 L. 13-16

M&M : P 6 L18: Here and the entire paper, including tables and figures; the terminology of saturated Fatty acids is not properly defined : the is one "0" to much 23:0 instead of 23:00 and so on for all the paper.

R: We have corrected the FA nomenclature through the paper.

P7 L4: I am puzzled by the transformation arcsine square root because % data are "transformed " (total 100%) which means that they have to be used as it is.

R: Proportional fatty acid (percentage of total FAME) data require transformation to meet the assumption of multivariate normality (Budge et al. 2006). The arcsine square root transformation is commonly used for proportional fatty acid data (e.g. Iverson, 2009; Thiemann et al., 2011; Raymond et al., 2014).

Budge, Suzanne M., Sara J. Iverson, and Heather N. Koopman. "Studying trophic ecology in marine ecosystems using fatty acids: a primer on analysis and interpretation." Marine Mammal Science 22.4 (2006): 759-801.

Iverson, Sara J. "Tracing aquatic food webs using fatty acids: from qualitative indicators to quantitative determination." Lipids in Aquatic Ecosystems. Springer New York, 2009. 281-308.

Thiemann, Gregory W., et al. "Individual patterns of prey selection and dietary specialization in an Arctic marine carnivore." Oikos 120.10 (2011): 1469-1478.

Raymond, Wendel W., Alexander T. Lowe, and Aaron WE Galloway. "Degradation state of algal diets affects fatty acid composition but not size of red urchin gonads." Marine Ecology Progress Series 509 (2014): 213-225.

P7 L10 : using SIMPER to identify potential FAs is somehow wrong , SIMPER give you what are the FA that contribute the most to the similarity . A small contribution of a "specific" FA, say a Branched one for Bacteria, would be a enough to trace the OM and still, will not show up in the best five of the primer analysis. This practice adds confusion on the data that are % but discussed in a quantitative manner.

R: SIMPER is a common routine to identify fatty acids that contribute to similarity within groups and dissimilarity between groups. We agree with the reviewer that this approach can be misleading in some cases. However, in this study, we used SIMPER only to identify the main fatty acids that characterized the primary producers sampled in the area (accordingly to Kelly and Scheibling, 2012). Then, the identified FAs were used as indicators of specific primary producer-derived organic matter in the abiotic compartments, assuming that the relative abundance of specific FA indicators in sedimentary or suspended organic matter will be proportional to the contribution of the correspondent primary producer.

The contribution of other potential organic matter sources to SOM and SPOM, as Bacteria, was assessed using the biomarkers published in literature.

To clarify better the aims and the results of this statistical approach, we have specified better the objectives of the SIMPER analysis highlighting that this approach is used to identify the FAs that contributed more to the similarity within and dissimilarity between primary producer groups (Clarke and Warwick, 2001). Moreover, we have specified that these FAs are used only as indicators of specific primary producer-derived organic matter for sedimentary and suspended particulate material characterisation, together with those reported in current literature (P. 7 L. 27 – P. 8 L. 3). To add clarity to these information, we have included also the dissimilarities between groups in the table (Suppl. 1).

Kelly, Jennifer R., and Robert E. Scheibling. "Fatty acids as dietary tracers in benthic food webs." Marine Ecology Progress Series 446 (2012): 1-22.

Clarke K.R., Warwick R.M. "Change in marine communities: an approach to statistical analysis and interpretation." Plymouth, UK: Primer-E (2001).

**Results :**
P7 L15to L21: all comparisons need to be tested statistically

R: All comparisons have been tested through permutational analysis of variance (PERMANOVA).

P8 L5 the Bayesian model (SIAR) is may be not needed to see the contributions of the sources since there is no fractionation to correct.

R: Bayesian models are valid tools to assess the contribution of sources to consumers (in this case SOM and SPOM) even regardless of fractionation value. In this study, we analysed the contribution of organic matter sources to sediment and particulate compartment assuming that their isotopic compositions remained unchanged after their incorporation in SOM and SPOM accordingly to Gonneea et al. (2004). Now have specified this in the manuscript. P. 7 L. 18-20.

P9 : L 25: Using 18:2w6+18:3w3 as tracers of seagrasses in zone full of mangrove is very risky.

R: We agree and now we were much more cautious in using FAs as tracers of primary producers. The whole paragraph has been rephrased, considering 18:3 n3 as a tracer of both seagrasses and mangroves, 18:2 n6 as a tracer of both seagrasses and agricultural runoff and 20:5 n3 as a tracer of both diatoms and red macroalgae. P. 11 L. 1-10

**Discussion** :
P10 L26 : In this paragraph it will be useful to discuss possible ocean inputs (seaward station)

R: The influence of oceanic input in the seaward station, in terms of influence of oceanic dissolved inorganic carbon on the carbon isotopic signature of primary producers, was discussed in the subsequent sentence P. 12 L. 14-18. In particular, we stated that "A similar enrichment was already observed in Gazi Bay and other tropical areas (Hemminga et al., 1994; Lugendo et al., 2007) and mirrors changes in $d^{13}C_{DIC}$ (Alongi, 2014; Maher et al., 2013). $d^{13}C_{DIC}$ is typically more negative close to mangroves as a result of the intense localized

mineralization of mangrove detritus (Bouillon et al., 2007) and increases seaward due to the increased contribution of oceanic DIC, whose d$^{13}$C is typically around 0‰ (Bouillon et al., 2008)". If necessary, further details will be added.

P13 L4-L5 : Speculative.

R: We will remove this sentence.

P4L15 : 16:1w7/16:0 is certainly not an indicator of dino/diatoms and , 20:5w3/22ww3 is a diatom/dino marker (not the opposite) . Another reason to not ascribed 20:5w3 to red algae.

R: We thank the reviewer for pointing out this mistake. Now we have corrected it and changed the sentence giving more importance to 20:5 n3/22:6 n3 as a diatom/dinoflagellate marker. We have decided to eliminate the mention to 16:1w7/16:0 because there was not a univocal pattern. P. 16 l. 2-4

P14 L25 : it is very speculative to relate a relative increase of bacterial FA (compare to what ? ) to an increase bacterial activity , at best it may show an increase in biomass but only if to compare the same site, for instance between season.

R: We are grateful to the reviewer for this comment. We will change "greater benthic mineralization" with "high bacterial biomass". p. 16  L. 10-12

Figure 1 : the limit of the mangrove forest is not obvious in the map, please write Mkurumunji RIVER

R: We agree that Figure 1 was not clear, as it was pointing out also by reviewer 1.
Then we have changed it to improve clarity. The names of the stations have been placed in the right position. "Creek" and "river" have been added to the watercourse names. Moreover, we have drawn the transects and the coastline to improve the identification of the limits of the mangrove forest.

Fig 7 and 8 : besides all my comments on the use of markers, here I would like to emphasize that Branched and 18:1w7 are surely tracers of bacteria, but one should complete the other and must not any more be added as it was done 20 years ago . there are many papers that show discrepancy between these two type of markers.

R: We have modified figures 7 and 8 (now 6 and 7) showing individually the two bacterial markers (branched and 18:1 n7) . Moreover, following previous comments of the reviewer, also 18:3 n3 and 18:2 n6 have been indicated individually.

[revised manuscript text omitted]